# Molecular basis of the inositol deacylase PGAP1 involved in quality control of GPI-AP biogenesis

Jingjing Hong[1,3], Tingting Li [1,3], Yulin Chao[2,3], Yidan Xu [1], Zhini Zhu[2], Zixuan Zhou[2], Weijie Gu[1], Qianhui Qu [2] ✉ & Dianfan Li [1] ✉

The secretion and quality control of glycosylphosphatidylinositol-anchored proteins (GPI-APs) necessitates post-attachment remodeling initiated by the evolutionarily conserved PGAP1, which deacylates the inositol in nascent GPI-APs. Impairment of PGAP1 activity leads to developmental diseases in humans and fatality and infertility in animals. Here, we present three PGAP1 structures (2.66–2.84 Å), revealing its 10-transmembrane architecture and product-enzyme interaction details. PGAP1 holds GPI-AP acyl chains in an optimally organized, guitar-shaped cavity with apparent energetic penalties from hydrophobic-hydrophilic mismatches. However, abundant glycan-mediated interactions in the lumen counterbalance these repulsions, likely conferring substrate fidelity and preventing off-target hydrolysis of bulk membrane lipids. Structural and biochemical analyses uncover a serine hydrolase-type catalysis with atypical features and imply mechanisms for substrate entrance and product release involving a drawing compass movement of GPI-APs. Our findings advance the mechanistic understanding of GPI-AP remodeling.

GPI anchoring of proteins is a crucial and ubiquitous posttranslational modification in eukaryotic cells (reviewed in refs. [1–4]). In mammals, GPI-APs predominantly reside in lipid rafts, playing essential roles in signaling, regulation, catalysis, and adhesion[2]. These functions are vital for development, immunity, and fertilization[2]. Genetic defects in GPI-AP synthesis can lead to severe inborn neurodevelopmental disorders[2], while downregulated GPI-AP remodeling is associated with postnatal neurodegenerative diseases[5–7].

In fungi, GPI-APs constitute a significant component of the cell wall[4], and in *Trypanosoma brucei*, the GPI-anchored variant surface glycoprotein (VSG) shields the parasite from the host immune system[8]. Therefore, enzymes involved in GPI-AP biogenesis present attractive targets for antimicrobial drug development[9–14]. Manogepix, a

compound targeting the GPI inositol acylation step, has progressed into phase 3 clinical settings[15] to treat fungal infections.

The GPI-AP biogenesis pathway is highly conserved, with enzymes from humans and *Saccharomyces cerevisiae* often being functionally interchangeable[2,4]. The pathway consists of over 20 catalysis steps divided into two phases (Fig. S1a–c) (reviewed in ref. [16]). During the synthesis phase, phosphatidylinositol (PI) undergoes stepwise modifications by an acyl group, mannoses (Man), and ethanolamine phosphates (EtNP) to form GPI, which is then attached *en bloc* to client proteins in the endoplasmic reticulum (ER) (Fig. S1a). In the post-attachment phase, GPI-APs undergo imperative lipid and glycan remodeling, including inositol deacylation and alternation of acyl type and saturation (Fig. S1b, S1c)[2,4,17]. These remodeling steps, along with

[1]State Key Laboratory of Molecular Biology, Center for Excellence in Molecular Cell Science, Shanghai Institute of Biochemistry and Cell Biology, Chinese Academy of Sciences; University of Chinese Academy of Sciences, 320 Yueyang Road, Shanghai 200031, China. [2]Shanghai Stomatological Hospital, School of Stomatology, Shanghai Key Laboratory of Medical Epigenetics, International Co-laboratory of Medical Epigenetics and Metabolism (Ministry of Science and Technology), Institutes of Biomedical Sciences, Department of Systems Biology for Medicine, Fudan University, Shanghai 200032, China. [3]These authors contributed equally: Jingjing Hong, Tingting Li, Yulin Chao. ✉e-mail: qqh@fudan.edu.cn; dianfan.li@sibcb.ac.cn

other quality control mechanisms such as the calnexin cycle[18,19] and cargo sorting[20], serve as crucial checkpoints for the proper sorting and transportation of GPI-APs.

The human PGAP1 (hPGAP1) or its yeast homolog Bst1 (referred to as yPGAP1 hereafter for consistency) initiates the post-attachment remodeling phase of GPI-AP biogenesis[21]. Residing in the ER membrane, PGAP1 removes the inositol-linked acyl chain from nascent triacylated GPI-APs (dubbed GPI-AP$_3$) to form the diacylated forms (GPI-AP$_2$). This process triggers the sorting and secretion processes that determine the fate of GPI-APs (note, most GPI anchors in mammalian cells contain 1-alkyl-2-acyl[2], but we use the term tri/diacyl unless specified otherwise). In *S. cerevisiae*, fluorescence microscopy studies have demonstrated that yPGAP1 is essential for sorting GPI-APs to specialized ER exit sites (ERES) distinct from that used by transmembrane proteins[22,23]. Therefore, the loss of yPGAP1 function leads to delayed ER export of GPI-APs such as Gas1[21,24,25]. Similarly, in mammalian cells, the deacylation by hPGAP1 is necessary for efficient ER export[21]. Mechanistically, the deacylation is required for binding with p24-family cargo receptors[22], which are topologically essential for recruiting GPI-APs into COPII vesicles (Fig. S1d)[26].

PGAP1 also plays a crucial role in the quality control of GPI-APs. In *S. cerevisiae*, loss of Bst1 (bypass sec thirteen 1) or other GPI-AP remodelases and p24 proteins[2] suppresses the lethality of the *sec13-1* mutant that is unable to assemble Sec13-mediated Coat Protein Complex II (COPII) vesicles for controlled GPI-AP transport at restrictive temperatures[27]. This bypass function comes at the expense of the fidelity of ER-to-Golgi transport, resulting in a reduced transport of GPI-APs and leakage secretion of ER-retained proteins. The uncontrolled transport is thought to be mediated by bulk flow vesicles[28], which are inhibited by PGAP1[27]. Further, PGAP1 interacts with the glycan quality control system through its interaction with calnexin[18,19], which binds the *N*-glycans of GPI-APs, thereby increasing their retention time in the ER for efficient deacylation (Fig. S1d). This collaborative process marks GPI-APs with fate-determining glycan and lipid codes for further sorting. Additionally, PGAP1 participates in the quality control of GPI-APs by facilitating their degradation through two types of mechanisms. Although GPI-APs are generally poor ERAD (ER-associated degradation) substrates[29], PGAP1 associates with misfolded GPI-APs and promotes their ERAD (Fig. S1e)[30]. In the second type, the degradation of misfolded GPI-APs in the lysosome involves secretion, including the RESET (rapid ER stress-induced export) pathway originally discovered for degradation of prions in mammal cells[31], microautophagy[32], or other endo-lysosome proteolysis pathways[29] (Fig. S1f). Finally, the cellular level of hPGAP1 is regulated by ERAD, and this regulation, mediated by the recently identified lipid scramblase TMEM41B, affects GPI-AP remodeling[33].

The physiological importance of PGAP1 has been described for all kingdoms of eukaryotic life. In humans, genetic mutations of PGAP1 are associated with global developmental delay[34], psychomotor retardation[35], intellectual disability[36], facial dysmorphism[37], and encephalophathy[38] characterized by impaired maturation of GPI-APs[36]. Further, schizophrenia patients exhibit dysfunction in ER export of several neural GPI-APs in the frontal cortex, which correlates with the down-regulation of PGAP1 and p24-family proteins[5]. Downregulation of PGAP1 also associates with Extramammary Paget disease[6]. In a mouse *pgap1* mutant named *beaker*, the animals display holoprosencephaly associated with down-regulated Cripto (a GPI-AP) signaling[39]. Another mutant called *oto* shows otocephaly due to up-regulated Wnt signaling[40]. Knockout (KO) of *PGAP1* in C57BL/6 mice is lethal in most cases, and male survivors are infertile due to the inability of sperms to migrate from the uterus to the oviduct and attach to egg cells[41]. Mechanistically, the loss of PGAP1 function may affect the proper localization of surface GPI-APs, such as lipid raft association, or their controlled shedding owing to the resistance of GPI-APs to GPI-specific phospholipases. These events are crucial in signaling/

developmental pathways and the fertilization process[42]. In plants, PGAP1 is required for efficient transport of GPI-APs and self-incompatibility (inability to produce zygotes after self-pollination)[43]. In *Candida albicans*, PGAP1 is critical for host invasion and immune escape[44]. Finally, the knockdown of PGAP1 in the bloodstream form of *Trypanosoma brucei* causes morphological abnormalities and decreases the surface expression of VSG[45], a GPI-AP responsible for escaping immune surveillance.

Despite the physiological importance and research advances in cell biology aspects of PGAP1, the biochemical properties and structural mechanisms of this unique lipase are not well understood. In fact, among the entire GPI-AP biogenesis pathway, only one enzyme has been recently structurally characterized[46,47], and to date, no enzymes have been kinetically studied, partly due to the challenges in obtaining the hydrophobic enzymes and chemically sophisticated lipid or lipidated protein substrates.

Here, we address this gap by developing a convenient assay using a fluorescent GPI-AP substrate and report the biological, biochemical, and structural characterizations of a fungal PGAP1, revealing its 10-transmembrane architecture with a lipase domain and a jelly-roll domain. Additionally, we successfully capture PGAP1 mutants in complex with a GPI-AP$_2$ and a fatty acid through cell-engineering strategies. By analyzing the product-bound structures and performing rational mutagenesis, we uncover mechanisms governing substrate fidelity, catalysis, and substrate entrance and product release. Our work realizes the kinetic characterization of a GPI-AP biocatalyst, provides clear visualization of a GPI-AP, offers a mechanistic understanding of the deacylation process, and provides a framework to design PGAP1 inhibitors as potential antimicrobial agents.

## Results

### Identification of a fungal PGAP1 that complements human and yeast PGAP1

In our quest to find stable proteins suitable for biochemical and structural studies, we identified PGAP1 from *Chaetomium thermophilum* (cPGAP1), which shares 20.7%/29.1% sequence identity and 33.8%/45.7% similarity with the human/yeast deacylase, respectively (Fig. S2a). To ascertain the functionality of cPGAP1, we conducted complementation assays in both human and yeast cells using full-length cPGAP1 constructs.

In HEK293 cells, the disruption of *PGAP1* renders the surface GPI-APs insensitive to phosphatidylinositol-specific phospholipase C (PI-PLC) due to the remaining inositol acyl[21] (Figs. 1a, S3a). Thus, the PI-PLC sensitivity of surface GPI-AP staining, which can be monitored by fluorescence-activated cell sorting (FACS), serves as an indicator for PGAP1 activity (Fig. S3b). As shown in Fig. 1b, overexpression of cPGAP1 restored the PI-PLC sensitivity of CD59 (a GPI-AP) in PGAP1-KO cells, indicating cross-species functional complementation, as previously reported for the yeast/human orthologs[21].

We also assessed the function of cPGAP1 in *S. cerevisiae*, a model system to study GPI-AP synthesis. The yeast *sec13-1* allele carrying the S224N mutation (previously[48] misannotated as S224K) of Sec13 is temperature sensitive due to the destabilizing mutation in this essential COPII coat protein. This lethal phenotype can be alleviated by the loss-of-function mutation of *PGAP1*[27] or other genes involved in the GPI-AP remodeling[28] and ER-Golgi transport[49]. Therefore, the lethality of *sec13-1* cells expressing PGAP1 constructs to be tested can indicate their inositol deacylation activity. Consistent with previous findings, *sec13-1* cells generated in this study exhibited temperature sensitivity (blue, Fig. 1c), and the knockout of yPGAP1 alleviated this lethal phenotype (magenta, Fig. 1c). Genomic integration of *cPGAP1* into the *sec13-1* yPGAP1-KO cells restored the temperature sensitivity (orange, Fig. 1c), similar to cells integrated with *yPGAP1* (green, Fig. 1c). These results demonstrate that cPGAP1 functions as a homolog of both the human and yeast PGAP1.

## Developing a fluorescence-based assay for biochemical characterization

The lack of commercial GPI or GPI-AP substrates has hindered enzymatic studies of the GPI-AP biogenesis pathway. In addition, the analysis of GPI-containing substrates/products largely relies on semi-quantitative methods such as thin-layer chromatography or SDS-PAGE.

To overcome the substrate availability challenge, we constructed a fluorescent GPI-AP by grafting the N- and C-terminal signal peptide sequence of decay-accelerating factor (DAF) onto Strep/Flag-tagged thermostable green fluorescence protein (TGP) (Fig. S4a). The chimera GPI-AP was sensitive to PI-PLC in a PGAP1-dependent manner in the FACS assay (Fig. S4b), suggesting it underwent GPI-anchoring and remodeling.

To produce this chimera GPI-AP with the inositol acyl, we over-expressed it in the HEK293 hPGAP1-KO cells. For simplicity, we refer to the GPI-AP version of TGP with triacyl, diacyl, and without acyl chains as $TGP_3$, $TGP_2$, and $TGP_0$, respectively. $TGP_3$ was solubilized in dode-cylmaltose and purified using Strep Tactin-affinity and size exclusion chromatography with a yield (2 mg per liter of culture) feasible for biochemical assays (Fig. S4c).

To overcome the semi-quantitative issue of conventional methods, we designed a fluorescence-detection size-exclusion chromatography (FSEC)-based coupled assay by taking advantage of the quantitative HPLC (high-performance liquid chromatography) system and the brightness of the TGP fluorescence. Principally, PGAP1 converts $TGP_3$ to $TGP_2$, which is converted to $TGP_0$ by excess amounts of PI-PLC (Fig. 1a). Both $TGP_2$ and $TGP_3$ (~30 kDa) carry acyl chains and should thus partition into detergent micelles (~70 kDa), reducing their retention time in size exclusion chromatography. This change should separate the PGAP1 substrate ($TGP_3$) from the final product ($TGP_0$) on FSEC, which is sensitive, reproducible, and convenient.

To prove this principle, $TGP_3$, $TGP_2$ (produced in wildtype HEK293 cells), and free TGP were analyzed on FSEC with detergents in the running buffer. As designed, $TGP_3$ and $TGP_2$ were eluted earlier than the free TGP (Fig. S4d).

Next, we performed the coupled reaction. Consistent with the design, a peak with similar $V_e$ of free TGP, which was presumed to be from the product ($TGP_0$) of the coupled reaction (Fig. 1a), emerged in a cPGAP1-, PI-PLC-, and dose-dependent manner (Fig. 1d). To validate this assay, we conducted the coupled assay using the existing phase separation method[21] that exploits the temperature-sensitive clouding

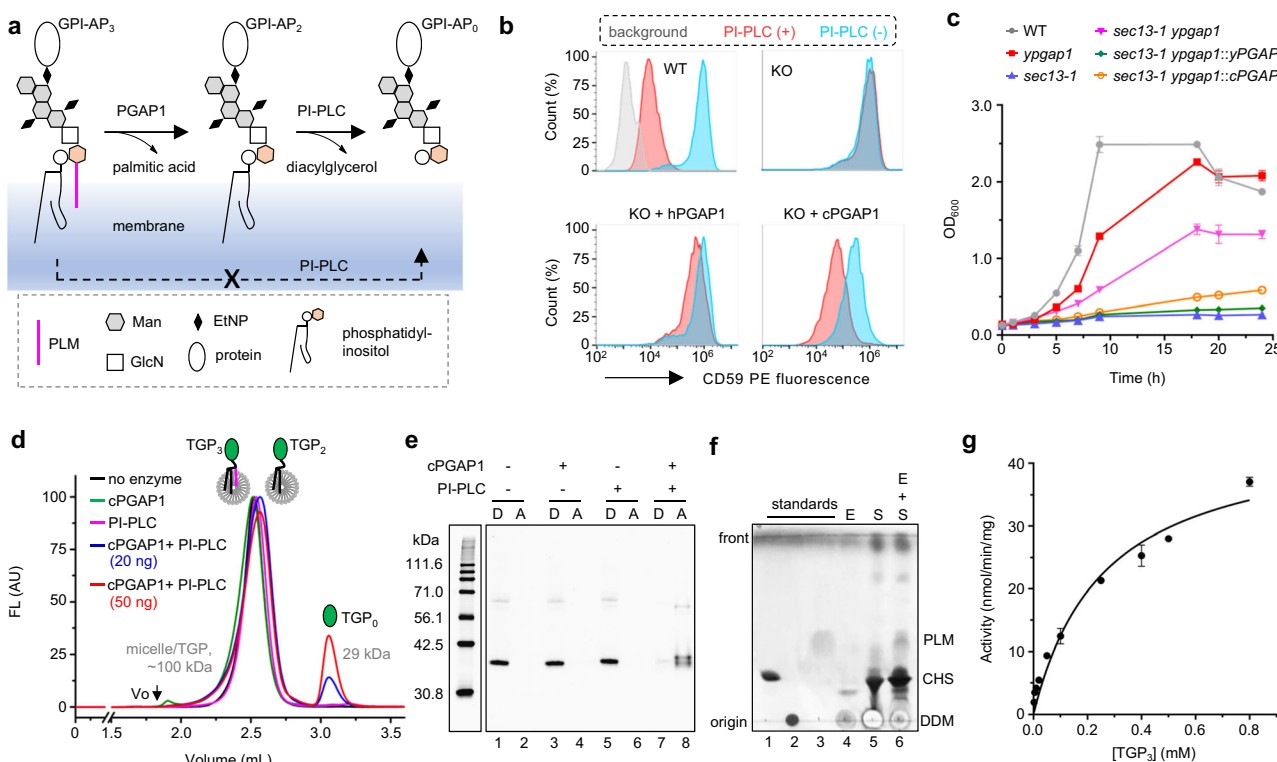

**Fig. 1 | Biological and biochemical characterization of cPGAP1. a** GPI-AP3 can only be converted to the water-soluble GPI-AP0 by the coupled reaction. **b** Expressing hPGAP1 and cPGAP1 (bottom) in HEK293 hPGAP1-KO cells (KO) restores PI-PLC sensitivity of CD59 staining in the FACS assay. Cells treated with (orange) or without PI-PLC (cyan) were analyzed by flow cytometry using gating strategies provided in Fig. S12a. PIGK-KO cells, in which GPI-AP biosynthesis are blocked[46,57], are used as a staining control (gray). In our experiments (n = 3 independent experiments), hPGAP1 reproducibly showed less activity than cPGAP1. **c** Integrating *cPGAP1* (orange) or *yPGAP1* (green) into *sec13-1* yPGAP1-KO cells (magenta) restores its temperature sensitivity as observed in the *sec13-1* cells (blue). Shown are the growth curve at 35 °C. These cells grew similarly at the permissive temperature (24 °C). yPGAP1-KO (red) in the wildtype cells (WT, gray) had little effect on cell growth. Genotype of the cells are indicated. Average ± s.e.m. (n = 3 independent experiments) are plotted. **d** FSEC of $TGP_3$ treated with cPGAP1 (green), PI-PLC (magenta), PI-PLC plus 20 ng (blue) or 50 ng (red) cPGAP1, or without enzymes (black). A representative of three independent experiments is shown. **e** Triton X-114-based phase separation assay. The detergent (D) and aqueous phase (A) of $TGP_3$ samples treated with indicated enzymes were analyzed by SDS-PAGE and in-gel fluorescence. The molecular weight of home-made fluorescence markers[81] is shown on the left. Representative results of three independent experiments are shown. **f** TLC assay. Lane 1-3 are cholesteryl hemisuccinate (CHS), dodecylmaltoside (DDM), and PLM standards. Reaction mix with the enzyme only (E, Lane4), substrate only (S, Lane 5), and with both (E + S, Lane 6) were extracted with chloroform/methanol and separated in a solvent containing hexane:diethyl ether:acetic acid (70:30:3, vol:vol). Typical results of three independent experiments are shown. **g** Michaelis-Menten kinetics of cPGAP1. Average ± s.d. (n = 3 independent experiments) are plotted. EtNP ethanolamine phosphate, FACS fluorescence-activated cell sorting, FSEC fluorescence-detection size exclusion chromatography, GlcN glucosamine, KO knockout, Man mannose, PI-PLC phosphatidylinositol-specific phospholipase C, PLM palmitic acid, TGP thermostable green fluorescence protein, TLC thin-layer chromatography. Source data of (**c**, **g**) and uncropped image for (**e**) are provided as a Source Data file.

of Triton X-114. In line with the FSEC results, the coupled reaction, but not the PGAP1 or PI-PLC reactions alone, converted $TGP_3$ from the detergent phase into the aqueous phase ($TGP_0$) (Fig. 1e). Additionally, in an orthogonal thin-layer chromatography assay, the cPGAP1 reaction produced fatty acids (Fig. 1f).

Using the FSEC assay, we determined the enzymatic kinetics parameters of cPGAP1. The activity of cPGAP1 was assayed at varying $TGP_3$ concentrations ranging from 2.5 to 800 μM (0.065−20.8 mg mL$^{-1}$). As shown in Fig. 1g, cPGAP1 displayed Michaelies-Menten kinetics, with a $V_{max}$ of 45 nmol min$^{-1}$ mg$^{-1}$ and $K_m$ of 262 μM.

### Structure determination of cPGAP1 in lipid nanodiscs and the product-bound complex in detergents

To obtain the cPGAP1 structure in a lipid environment, we purified the full-length cPGAP1 with a C-terminal TGP fusion in detergents and reconstituted it into lipid nanodiscs made of the membrane scaffold protein (MSP) 1E3 (average diameter of 12 nm)[50] and 1-palmitoyl-2-oleoyl-phosphatidylserine (POPS). cPGAP1 co-eluted with MSP with a monodisperse and symmetric peak on gel filtration (Fig. S4e), indicating successful reconstitution. Using single-particle cryo-electron microscopy (cryo-EM), we determined the structure of cPGAP1 at a nominal resolution of 2.83-Å (cPGAP1$^{apo}$, Fig. 2a, Fig. S5, Table S1). Similar to previously reported[51,52] high-resolution nanodisc structures, belt-like densities likely representing the amphipathic scaffolding helices of MSP that wrap the lipid bilayer were observed near the membrane boundaries.

To gain insights into substrate recognition and catalysis, we attempted to capture a structure with $TGP_3$ bound to cPGAP1 using catalytically impaired mutants. Among the three residues in the Ser327-His443-Asp407 triad (which we will discuss later), we initially selected the H443N mutant for two reasons. First, it exhibited only residual activity (0.4% of wildtype) in the FSEC-assay (Table S2). Second, compared to the S327A mutant, the asparagine replacement of a histidine residue in H443N was expected to cause fewer perturbations to the triad geometry and thus to ligand binding due to the similar volume and solubility between these two residues. To facilitate the expression of both the enzyme and the substrate, we tagged cPGAP1 with mCherry.

Since the yield by mixing the separately purified cPGAP1 with $TGP_3$ was low, we co-expressed Strep-tagged $TGP_3$ and His-tagged cPGAP1 H443N and purified the complex (dubbed cPGAP$^{H443N}$) using tandem affinity chromatography and gel filtration (Fig. S4f). The gel filtration profile monitoring absorbance of general proteins (280 nm) and enzyme/ligand (585 nm for PGAP1-mCherry, 493 nm for GPI-anchored TGP), as well as the in-gel fluorescence, indicated co-elution of cPGAP1$^{H443N}$ and $TGP_{2/3}$ (Fig. S4f). Subsequently, we determined the cryo-EM structure of the liganded complex to a nominal resolution of 2.68-Å (Fig. S6a−d).

The cryo-EM map reveals the presence of both the deacylase and a GPI-AP (Fig. 2b). However, the ligand density near the scissile bond was broken (Fig. 2c). Thus, instead of a triacylated substrate ($TGP_3$), the results supported the fitting of the two products, namely a fatty acid (palmitic acid, PLM) and a diacylated GPI-AP ($TGP_2$). In addition, the PLM carboxyl is distant (5.3 Å) from the inositol 2-OH, where it is attached in $TGP_3$ (Fig. 2c), suggesting the fatty acid had been hydrolyzed. Consistent with the structural observation, the sample used for cryo-EM grid preparation contained a significant portion of $TGP_2$, as demonstrated by its PI-PLC sensitivity in the FSEC assay (Fig. 2d). Moreover, cPGAP1 H443N exhibited noticeable cellular activity against CD59 in the FACS assay (Fig. 2e), despite its low activity (0.4% to the wildtype) in the FSEC assay (Table S2). This discrepancy likely reflects the limited substrate availability in relation to the overexpressed enzymes in the cell-based FACS assay. Taken together, the cPGAP1$^{H443N}$ structure represents the product-bound state. Next, we turned our

attention to cPGAP1 S327A. Unlike H443N, S327A did not show activity even with prolonged reaction time in our FSEC assay (Fig. S4g). In addition, hPGAP1 S174A, the human counterpart of cPGAP1 S327A, showed no activity in cell-based FACS assays[21]. Therefore, we co-expressed cPGAP1 S327A with $TGP_3$, purified the complex (Fig. S4h) similarly to cPGAP1 H443N, and determined its cryo-EM structure (dubbed cPGAP1$^{S327A}$) to a global resolution of 2.66 Å (Fig. S6e−h, Table S1). To our surprise, however, the density also supported the fitting of the products ($TGP_2$ and PLM) (Fig. 2f), rather than $TGP_3$, as similarly reasoned for cPGAP1$^{H443N}$ (Fig. 2c).

The unexpected hydrolysis of $TGP_3$ prompted us to investigate the activity of cPGAP1 S327A further. Consistent with the structural observation, a significant portion of $TGP_3$ was hydrolyzed to $TGP_2$ in the cryo-EM sample, as revealed by the FSEC assay (Fig. 2d). A retrospective FACS assay, however, showed no activity for S327A on the co-expressed $TGP_3$ (Fig. 2g), which is in line with previous reports[21]. The discrepant results indicated that the S327A mutant became active during purification. Indeed, $TGP_2$ production was evident in the cPGAP1 S327A-expressing cells, but not in the negative control cells expressing the double mutant S327A/H443N, after cell lysis (Fig. S4i). Based on these results, we concluded that cPGAP1$^{S327A}$ also exists in a product-bound state.

### Overall structure and domain arrangement

The three structures are overall similar, with a Cα RMSD (root-mean-square deviation) of 0.04 Å between the two product-bound structures and 0.38 Å between cPGAP1$^{apo}$ and the liganded structures. Due to its higher model completeness (81% of its 1183 residues, Fig. S7a), the cPGAP1$^{S327A}$ structure is used here for structural description. cPGAP1 contains a transmembrane domain (TMD) (Fig. 3a, b), a lipase domain with a typical α-β-α hydrolase architecture, and a third domain with two jelly-roll subdomains that show a classic β-sandwich fold (Figs. 3c, S7a). Together, they form a wrench shape, with the TMD as the handle, the lipase domain and the jelly-roll domain as the fixed and adjustable jaws, respectively (Figs. 2b, 3a).

The TMD creates a bowl-shaped crater at the luminal interface to host the lipase domain. Through this arrangement, these two domains form a seemingly stand-alone entity (Fig. 3a) to which the products are bound (Fig. 2b), suggesting that the jelly-roll domain is not directly involved in the catalytic activity of cPGAP1. The jelly-roll domain is similar in size to the lipase domain (Fig. 3a). The bottom subdomain is associated with the lipase domain, while the upper subdomain is separated from it, creating a crevice of approximately 20 Å × 20 Å × 10 Å in dimension (Fig. 3a). This geometry exposes a large surface area, which may be advantageous for weak interactions with its substrate GPI-APs that can vary in size and shape.

### Characteristics of each domain

The TMD consists of 10 transmembrane helices (TMHs) that are mostly perpendicular to the membrane, except that the long TMH4 (44 residues) crosses the membrane diagonally at an approximate 40° angle (Figs. 3a, S7a). All TMHs, except for TMH1 and TMH5, are tightly packed with TMH7 in the middle (Fig. 3b). TMH1 serves as a membrane anchor for Loop1, which connects TMD and the lipase domain. This connection, together with Loop1's patch-like interaction with the lipase domain and the jelly-roll domain, stabilizes cPGAP1. This stabilizing role is further emphasized by an evolutionarily conserved (Fig. S2b) disulfide bond between Cys177 of Loop1 and Cys449 of the lipase domain (Fig. 3d). Consistent with this notion, substituting the thiol group with a hydroxyl by the C177S mutation almost abolished activity (Fig. 3e, Table S2). This mutant also exhibited a lower purification yield compared to the wildtype (Table S2), suggesting misfolding or instability due to the loss of the disulfide bond.

The two jelly-roll subdomains, which also exist in the predicted[53] hPGAP1 and yPGAP1 models (Fig. S7b), pack against each other

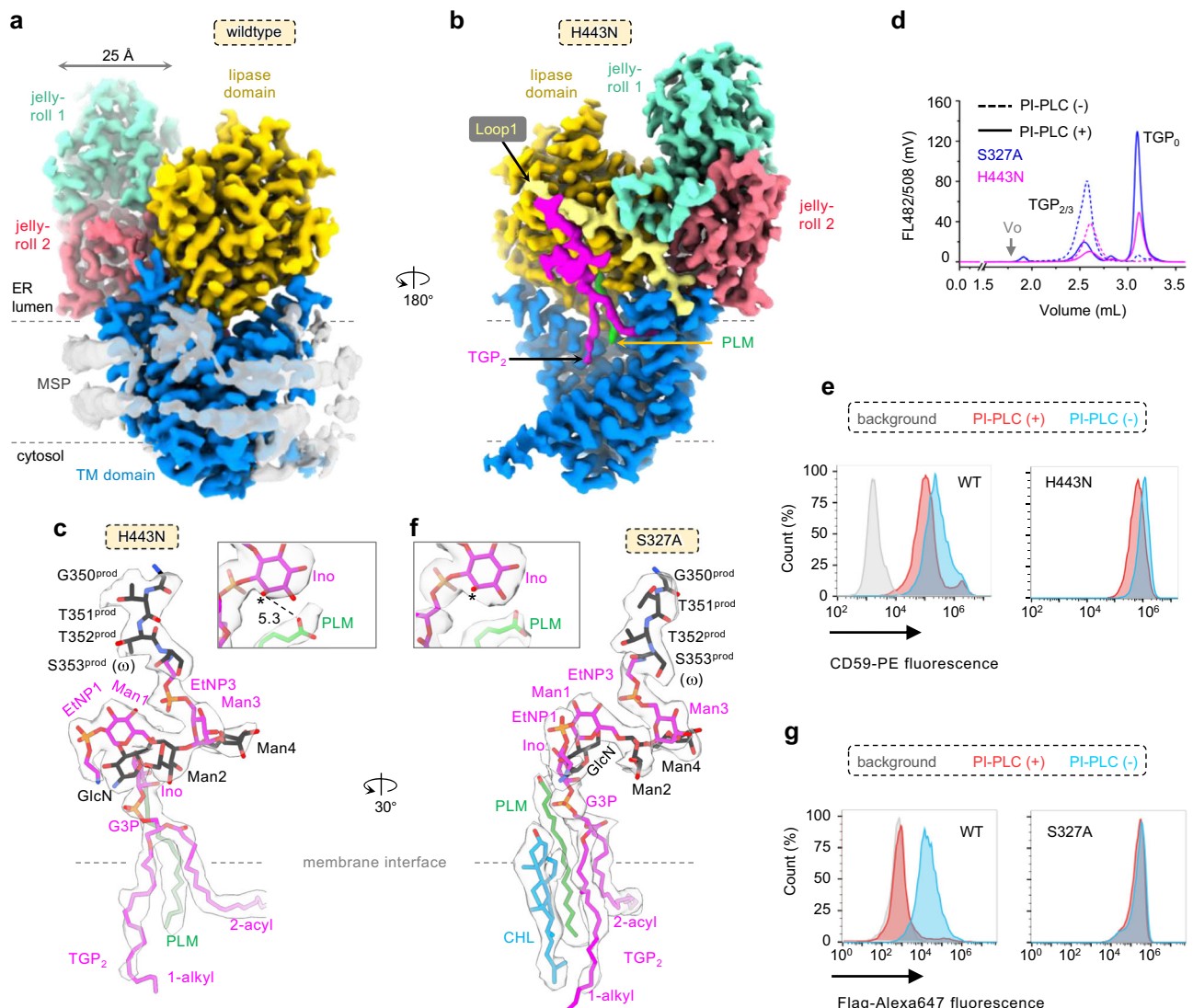

**Fig. 2 | Structural and functional characterization of cPGAP1 wildtype and mutants.** Cryo-EM maps of wildtype cPGAP1 (**a**, 2.84 Å) in lipid nanodiscs (transparent gray, color-coded by domains as indicated) and cPGAP1$^{H443N}$ (**b**, 2.68 Å) with TGP$_2$ (magenta) and PLM (green) bound. **c** Expanded view of the density (transparent gray) and model of the products in cPGAP1$^{H443N}$. An asterisk indicates the palmitoylated inositol 2-hydroxyl in GPI-AP$_3$. A dashed line indicates the distance (Å) between this hydroxyl and the carboxyl group. **d** FSEC profile (single experiment) of the cryo-EM sample for H443N (magenta) and S327A (blue), treated with (solid) or without (dash) PI-PLC. **e** FACS of hPGAP1-KO cells expressing cPGAP1 wildtype (WT) or H443N treated with (orange) or without (cyan) PI-PLC using CD59 as the marker with gating strategies provided in Fig. S12a. **f** Expanded view of the density (transparent gray) and model of the products and a nearby cholesterol molecule in cPGAP1$^{S327A}$. An asterisk indicates the palmitoylated inositol 2-hydroxyl in GPI-AP$_3$. In (**c** and **f**), TGP$_2$ is colored alternatingly for better visualization, PLM is colored green, and CHL is colored cyan. A superscript prod marks the protein residues of TGP$_2$. **g** FACS of hPGAP1-KO cells expressing cPGAP1 wild-type (WT) or S327A treated with (orange) or without (cyan) PI-PLC using the Flag-tag on TGP$_{2/3}$ as the marker with gating strategies provided in Fig. S12c. The rotation arrow in (**a**, **b**) and (**c**, **f**) indicate the view angle of different, superposed objects. In (**e** and **g**), (1) typical results of three independent experiments are shown; (2) PIGK-KO cells, in which GPI-AP biosynthesis are blocked[46,57], are used as a staining control (gray). CHL cholesterol, EM electron microscopy, EtNP ethanolamine phosphate, ER endoplasmic reticulum, FACS fluorescence-activated cell sorting, FSEC fluorescence-detection size exclusion chromatography, G3P glycerol 3-phosphate, GlcN glucosamine, GPI-AP glycosylphosphatidylinositol-anchored protein, Ino inositol, KO knockout, Man mannose, MSP membrane scaffolding protein, PI-PLC phosphatidylinositol-specific phospholipase C, PLM palmitic acid, TGP thermostable green fluorescent protein.

through a network of hydrophobic interactions, mostly of aromatic residues (Fig. 3c). Attempts to dissect the function of the two jelly-rolls by subdomain deletion were unsuccessful, as the deletion mutants (see Methods) did not express in HEK293 cells. The deletions likely expose the abovementioned hydrophobic patch, causing protein misfolding. Interestingly, although deleting both jelly-roll subdomains rendered the hPGAP1 inactive, the similar deletion construct of cPGAP1 exhibited deacylase activities in cells (Fig. 3f). Biochemical characterization of cPGAP1 Δjelly-roll was unsuccessful, however, as the protein precipitated during affinity chromatography despite showing the expected molecular weight on SDS-PAGE without noticeable

degradation (Fig. S8). Although the exact function of the jelly-roll domain remains to be studied in the future, our results indicate that it is not necessary for deacylase activity.

## A pocket with balanced interactions for substrate fidelity

A remarkable feature of GPI-AP biosynthesis is that a single set of enzymes processes numerous substrates. As the proproteins' main bodies vary significantly in size and shape, their processing relies on a C-terminal signal peptide, which only possesses a vague pattern rather than consensus sequences[2]. In the remodeling phase, even this vague pattern is mostly lost, as GPI had replaced the signal peptide[46]. The

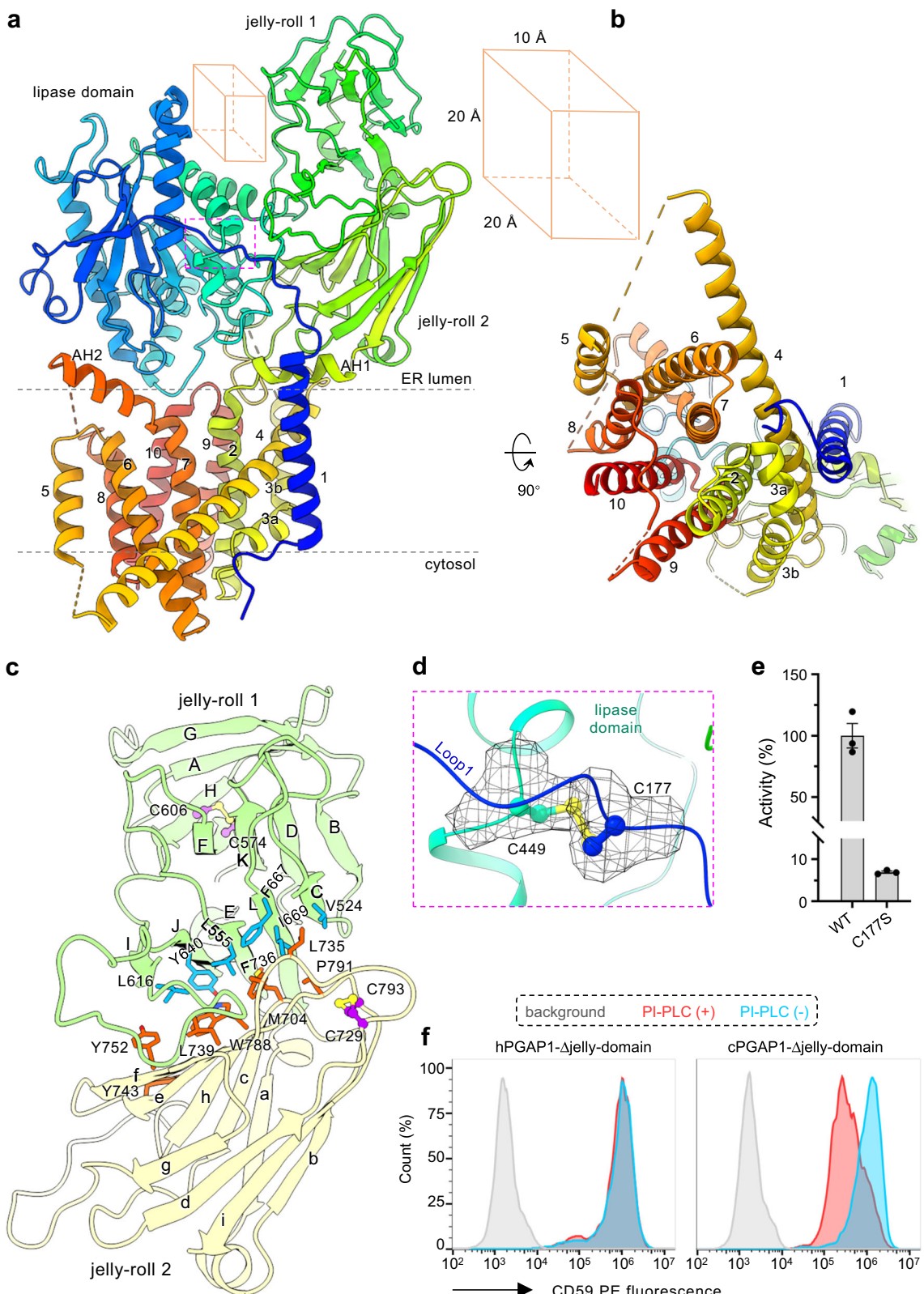

only common feature left is a flexible, 5–10 residue-long ω-minus region (N-terminal to the attachment ω-site). Therefore, the ω-minus region and the GPI anchor, but not the main body of GPI-APs, confer substrate specificity for GPI-AP remodeling enzymes.

In line with this, the main body of GPI-AP, in this case, the TGP, was not visible, while four ω-minus residues (Gly$^{350}$-Thr-Thr-Ser$^{353}$) were well resolved in the product-bound structures (Fig. 2c, f). Notably, the

substrate TGP$_3$ fitted into the density superimposes well with the two products (Fig. S9a, S9b), with the scissile bond near the catalytic Ser327 (Fig. S9c). Therefore, we use the product-bound structures to infer substrate recognition, although we acknowledge that the interactions with the catalytic residues would differ slightly between the two states. The two product-bound structures are overall similar, and cPGAP1$^{H443N}$ is used to describe the ligand-enzyme interactions.

**Fig. 3 | Structure of cPGAP1.** Overall structure of cPGAP1 in rainbow-colored cartoon representation viewed from the membrane plane (**a**) and cytosol (**b**). Numbers 1-10 indicate the ten transmembrane helices (TMHs), while 3a/3b marks the broken TMH3. Dashed lines indicate membrane boundaries calculated using the OPM server[82]. AH denotes amphipathic helix. A cubic box illustrates the dimension (Å) of the crevice between the lipase domain and the jelly-roll domain. **c** Expanded view of the jelly-roll domain (cartoon; jelly-roll 1, green; jelly-roll 2, yellow) with residues involved in the packing of the two subdomains highlighted (stick). Disulfide bond residues (purple) are shown as ball-and-stick representations. **d** Expanded view (the magenta box in **a**) of the inter-domain (Loop1, blue;

Lipase domain, cyan) disulfide bond (ball-and-stick). The Cryo-EM density of the cystine is shown as mesh. **e** Activity (% of wildtype, average ± s.e.m., $n = 3$ independent experiments) of the C177S mutant. **f** Flow cytometry of PGAP1-KO cells expressing hPGAP1 ΔJelly-roll or cPGAP1 ΔJelly-roll treated with (orange) or without (cyan) PI-PLC using CD59 as the GPI-AP reporter with gating strategies provided in Fig. S12a. PIGK-KO cells, in which GPI-AP biosynthesis are blocked[46,57], are used as a staining control (gray). Typical results of three independent experiments are shown. Source data of (**e**) are provided as a Source Data file. EM electron microscopy, KO knockout.

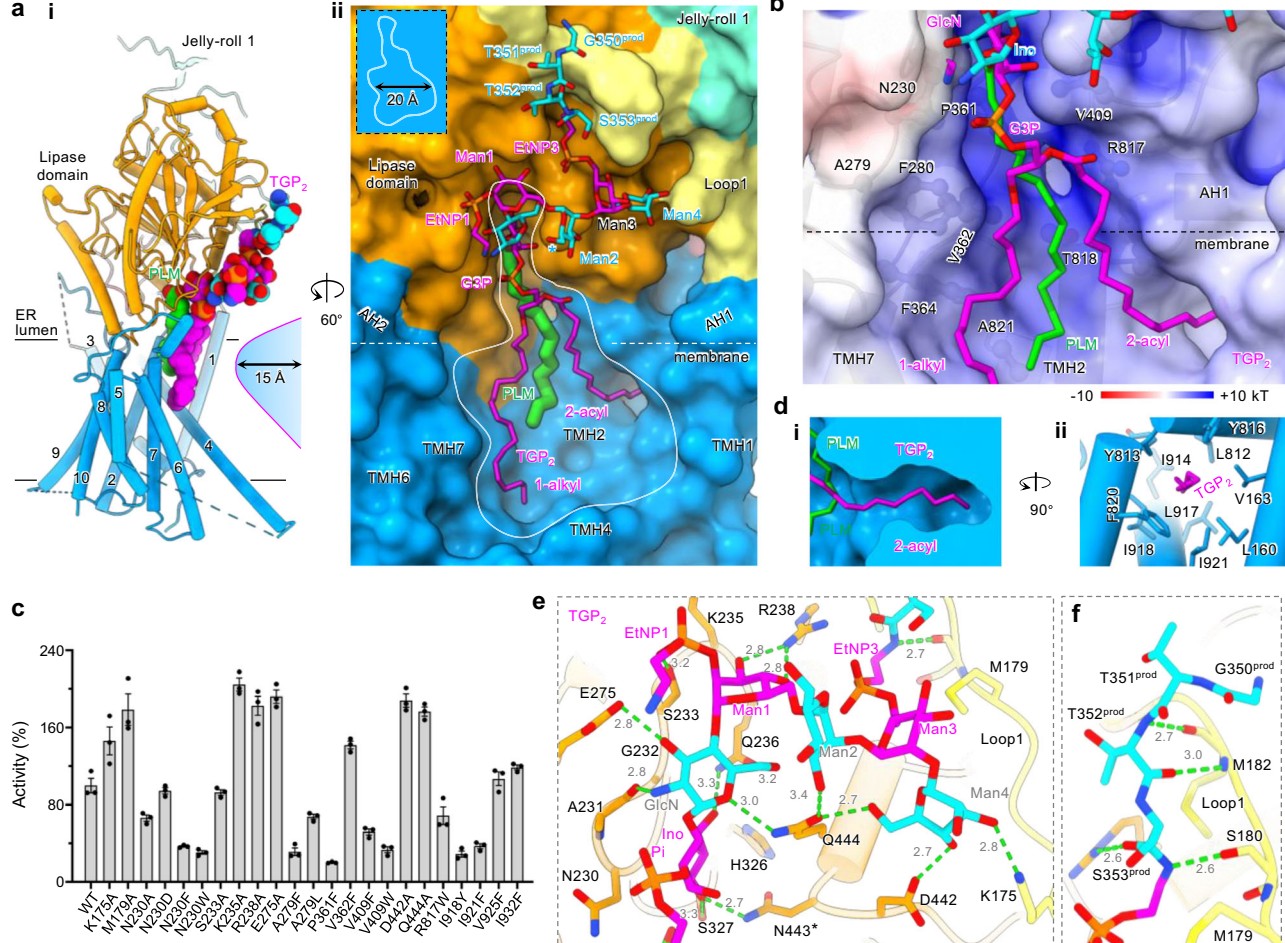

**Fig. 4 | Liganded structures inform mechanisms for substrate fidelity.**
**a** Overview (**i**) and expanded view (**ii**) of cPGAP1[H443N]. cPGAP1 (colored by domains, lipase domain, orange; loop1, light yellow; jelly-roll 1, cyan; TMH, marine) is shown as cartoon (**i**) or surface representations (**ii**). A white line (**ii**) outlines the guitar-shaped cavity. Palmitic acid (green) and TGP₂ (alternating magenta and cyan) are shown as stick representations. **b** Semi-transparent electrostatic potential surface of the lipid-binding cavity generated using the Adaptive Poisson-Boltzmann Solver[83] server. Residues lining the cavity are additionally shown in ball-and-stick representations. **c** Activity of cPGAP1 mutants. Activity from the FSEC-based assay is normalized to the wildtype

(WT) and plotted as average ± s.e.m. ($n = 3$ independent experiments). **d** The 2-acyl chain of TGP₂ (magenta stick) inserts into a deep pocket (clipped surface in **i**, cartoon and stick in **ii**) by 5-6 carbon atoms. The H-bonding network that holds the glycans (magenta and cyan) (**e**) and amino acid residues (cyan) (**f**) of TGP₂ to cPGAP1 (orange, lipase domain; yellow, Loop1). Green dashed lines denote H-bonds with distance (Å) indicated in numbers. An asterisk in Asn443 indicates the H443N mutation. AH amphipathic helix, ER endoplasmic reticulum, EtNP ethanolamine phosphate, G3P glycerol 3-phosphate, GlcN glucosamine, Ino inositol, Man mannose, PLM palmitic acid, TMH transmembrane helices, TGP thermostable green fluorescence protein. Source data of (**c**) are provided as a Source Data file.

The products bind cPGAP1 with a large buried surface area of 1641 Å². PLM and the phospholipid part of TGP₂ are hosted in a deep (~15 Å) lateral cavity formed jointly by the lipase domain and TMH1/4/6/7 (Fig. 4a). Remarkably, both TGP₂ and PLM are lifted towards the lumen (Fig. 4a) to an extent where half of PLM surpasses the calculated membrane interface. Moreover, an electropositive patch runs through the PLM cleft (Fig. 4b). These energetically unfavorable effects are likely counterbalanced by enzyme-glycan interactions in the luminal

space, a unique characteristic of GPI-AP that the enzyme exploits as a specificity filter to avoid hydrolyzing bulk phospholipids.

## Substrate recognition for the GPI acyl chains
The guitar-shaped cavity (Fig. 4a) is optimally organized to suit substrate and products-binding. The narrow fingerboard, lined by Asn230/Ala279/Phe280/Val362/Phe364 on one side and Val409/Arg817 on the other side, with Pro361/Thr818/Ala821 at the bottom (Fig. 4b),

constrains the PLM, while the spacious sound chamber region accommodates the diacyl chains of $TGP_2$.

Consistent with the structural observations, the N230A mutation reduced inositol deacylase activity to 66% of the wildtype (WT). In addition, while introducing bulky residues like N230F and N230W further reduced activity to 36% and 30%, respectively, substituting Asn230 with aspartate to maintain a similar sidechain volume had little effect on activity (Fig. 4c), suggesting the importance of the cavity shape in substrate binding. Similarly, replacing Ala279 with the bulkier leucine to narrow the PLM cleft reduced activity by ~33%, and introducing an even bulkier residue (A279F) further exacerbated the effect by an additional 36% (Fig. 4c). Similar trends were observed for residues on the other side of the cleft (V409F, 52%; V409W, 33%; R817W, 69%) and at the bottom (P361F, 20%).

In contrast to the constrained binding of the inositol acyl, the diradyl moiety has more freedom in the chamber, showing some level of flexibility as indicated by its less-defined density (Fig. 2c). Typically, in mammalian cells, the 2-acyl chain of $GPI\text{-}AP_3$ consists of unsaturated fatty acids with 16 to 22 carbons[54]. In the two product-bound structures, $TGP_2$ was built with a C18:0 ($cPGAP1^{H443N}$) and a C20:4 ($cPGAP1^{S327A}$) 2-acyl chain (see Methods) without causing clashes. Moreover, the chamber appeared spacious enough to accommodate a C22 2-acyl chain. Consistently, introducing bulky residues at several positions in the chamber (V362F, V925F, I932F) had no impact on or even increased the deacylase activity (Fig. 4c).

A notable feature of GPI-cPGAP1 binding is the insertion of the 2-acyl chain into a deep pocket (~10 Å) by 5−6 carbons (Figs. 4d, S10). Mutations intended to fill the pocket and block the insertion drastically impaired cPGAP1 activity (I918Y, 29%; I921F, 37%). Finally, the V163F mutant was severely degraded (Fig. S8), suggesting that disturbing this pocket leads to misfolding or protein instability.

## Substrate recognition for the GPI glycans

Adjacent to the phospholipid part, the inositol and glucosamine residues sit atop the fingerboard region in a tight cleft, which involves Asn230/Ala231/Gly232/Ser233 on one side and Ser327/Val409/Asn443 (His443 in the WT)/Gln444 on the other side, with Gln236/His326 at the bottom. Apart from shape complementation, seven H-bonds help secure the two glycan residues in place (Figs. 4e, S10).

Further up the glycan core, six H-bonds are formed between the mannoses and cPGAP1. These include two between Man1 and Arg238, one between Man2 and Gln444, and three between Man4 and Lys175/Asp442/Gln444 (Figs. 4e, S10). The greasy slide interaction, often observed between the hydrophobic face of sugar and aromatic protein residues[55], was not evident in our structures.

Regarding the decorating phosphates, EtNP1 and the bridging EtNP3 each form one H-bond with Ser233 and Ser180, respectively (Figs. 4e, S10). Interestingly, no electrostatic interactions were observed despite their zwitterionic nature and proximity to the electropositive Lys235 and Arg238. In fact, K235A and R238A even displayed higher activity than the WT (Fig. 4c, Table S2). Mechanistically, strong ionic interactions for enzyme-product binding may need to be avoided as they may impede the release of products.

Despite the well-defined density of the GPI-anchor, there was no evidence for the EtNP2. EtNP2 may exist in a highly mobile state due to a lack of interactions with cPGAP1, as suggested by the fact that the Man2 6-hydroxyl (where EtNP2 is attached) is exposed to the bulk solvents (Fig. 4a). This exposure implies accessibility to PGAP5, an ER membrane-residing remodelase that cleaves EtNP2 and primes GPI-APs for ER export[56]. Extending this idea, it is conceivable that EtNP2 may even have been removed, especially given the presumed long retention of $TGP_{2/3}$ in the ER membrane, attributed to the compromised activity of cPGAP1 H443N and cPGAP1 S327A.

In our previous studies[46,57] of GPI-T, the enzyme catalyzing the attachment step, residues forming H-bonds with the EtNP-decorated

mannoses were insensitive to single mutations, likely due to the rich network of interactions with the flexible glycans. This mutation-resistant interaction mode was also observed between GPI and cPGAP1. Thus, alanine mutations of Lys175/Met179/Ser233/Glu275/Asp442/Gln444 did not impact or even increased activity (Fig. 4c, Table S2).

## Substrate recognition for GPI-AP protein residues

As mentioned above, one of the hallmarks of the proproteins is a structurally flexible ω-minus region with no sequence consensus. Our structure rationalizes this flexibility. First, the four protein residues adopt an extended conformation with no ordered secondary structures (Fig. 4f). Second, the peptide resides in a shallow groove, with the sidechains protruding into the bulk solvents without specific interactions (hydrogen or ionic bonds) with the enzyme (Fig. 4f). Instead, they form β-sheet like backbone interactions with residues from cPGAP1 Loop1 (Fig. 4f). These structural characteristics allow broad substrate sequence specificity in the ω-minus region. Finally, based on the analyses above, the peptide-mediated interactions only make a small contribution to the overall GPI-AP binding (Fig. S10), supporting previous findings[58] that PGAP1 can effectively process free GPI molecules in cells.

## The catalytic mechanism

The lipase domain of cPGAP1 exhibits a typical α/β hydrolase fold (Fig. 5a), with an eight-stranded β-sheet sandwiched by α-helices. Superposing $cPGAP1^{H433N}$ onto $cPGAP^{apo}$ positions the PLM carboxylic acid close to the catalytic Ser327 (Fig. 5a), which forms the catalytic triad with the nearby His443 and Asp407. As mentioned, S327A showed no activity with separately purified $TGP_3$ as the substrate. Substituting Ser327 with cysteine drastically reduced activity (7% of WT) (Fig. 5b), while the H443N mutant displayed residual activity (0.4%) (Fig. 5b). For Asp407, the neutralizing mutant D407N reduced the activity to 22%, while trimming the sidechain to Cβ (D407A) reduced activity to 8% of WT (Fig. 5b).

Based on the structural analysis and mutagenesis data, we propose a mechanism similar to triad lipases and serine proteases[59] (Fig. 5c). Asp407 increases the $pK_a$ of His443, which in turn activates the Ser327 hydroxyl. On the other hand, the substrate $GPI\text{-}AP_3$, as illustrated in manually fitted and superimposed models in Fig. S9, is positioned with the scissile ester bond near the oxyanion hole, with the carboxyl of the superposed PLM and $GPI\text{-}AP_3$ sandwiched between the backbone amides of Met328 and Asn230 (Fig. 5a). Akin to triad hydrolyases[59], this configuration further polarizes and activates the C = O bond, facilitating the nucleophilic attack of the carbonyl carbon (of the manually fitted $GPI\text{-}AP_3$) by the primed Ser327 (Fig. 5c, i), which is 3.0-Å away (measured from manually fitted and superimposed models) (Fig. 5a). This results in the formation of an enzyme-substrate ether bond at the expense of the collapse of the carbonyl (Fig. 5c, ii). The product $GPI\text{-}AP_2$ and an acylated intermediate are generated through the regeneration of the carbonyl (Fig. 5c, ii). Subsequently, a water molecule activated by His443 attacks (Fig. 5c, iii) and hydrolyzes the intermediate through another cycle of collapse and regeneration of the carbonyl moiety (Fig. 5c, iv), producing the product fatty acid (FA) (Fig. 5c, v).

## A proposed mechanism for substrate entrance and product release

Despite the overall structure similarity, the two product-bound structures ($cPGAP1^{H443N}$ and $cPGAP1^{S327A}$) exhibit notable differences in the positioning of the $TGP_2$ sn-1 alkyl chain, offering insights into the processes of substrate entrance and product release. In $cPGAP1^{H433N}$, the 1-alkyl chain tightly associates with the enzyme (Fig. 5d). Conversely, in $cPGAP1^{S327A}$, it undergoes a ~45° drawing compass rotation, shifting outward by 6 Å and detaching from the enzyme (Fig. 5d). This

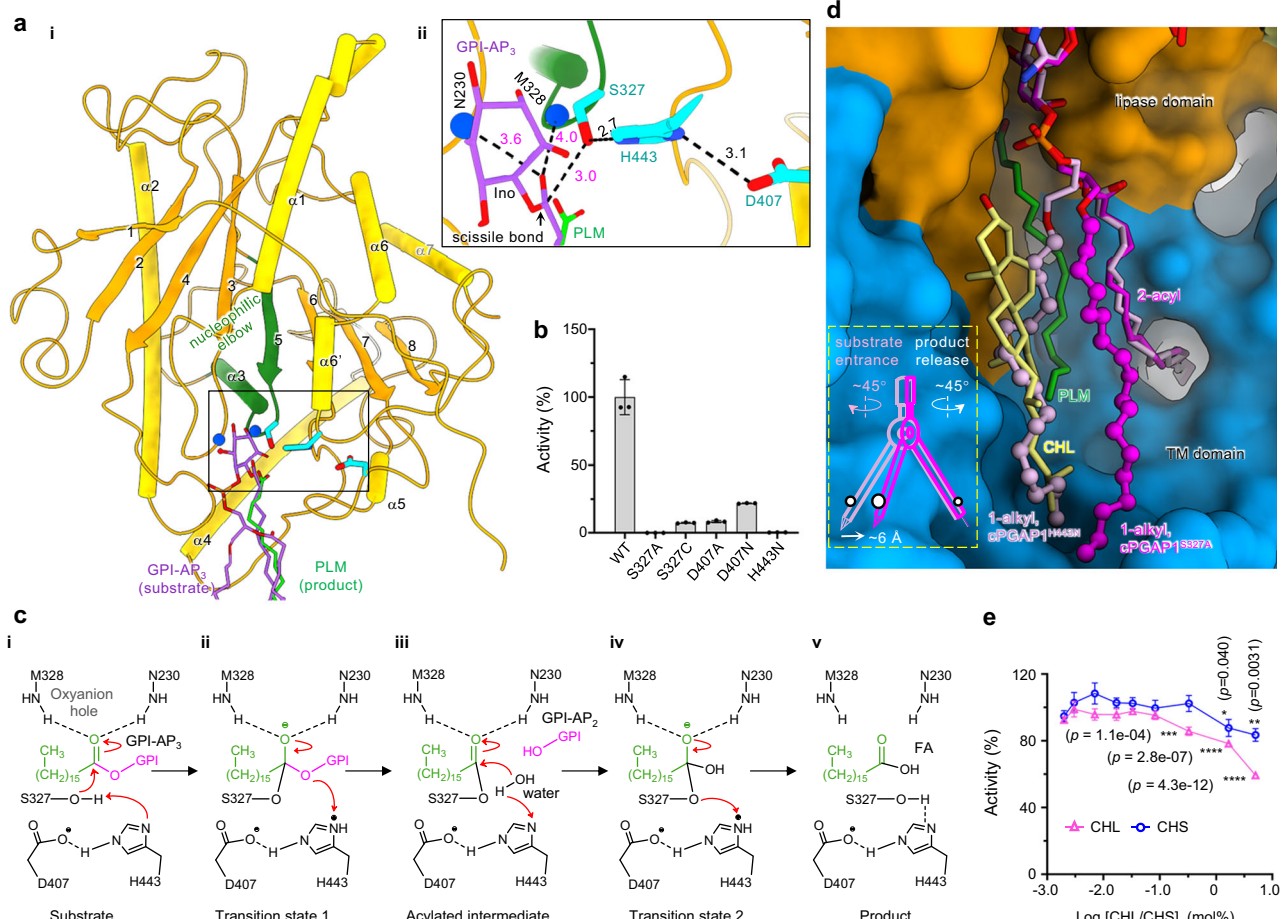

**Fig. 5 | Mechanisms for catalysis and product release. a** Overview of the lipase domain (**i**) and expanded view of the catalytic site (**ii**) of cPGAP1$^{apo}$. Elements including the central β-sheet (strands 1-8) (orange), the sandwiching α-helices (yellow), the nucleophilic elbow (green), the catalytic triad (cyan), the backbone nitrogen of the oxyanion hole (blue) are highlighted. Palmitic acid (PLM, green), and the manually fitted GPI-AP$_3$ (only showing relevant parts, purple) (Fig. S9) are superimposed to illustrate the proposed substrate positioning. Numbers indicate distances (Å) either observed structurally (black) or measured from superposed ligands (magenta). **b** Activity (% to wildtype) of triad mutants. Average and s.e.m. ($n = 3$ independent experiments) from three FSEC assays are plotted. **c** The serine hydrolase-type mechanism. The structures from this work only captured the state **v**. The rest of the steps were speculated based on existing knowledge on triad proteases. **d** A proposed mechanism for substrate entrance and product release involving a drawing compass movement (inset) of TGP$_2$ (cPGAP1$^{S327A}$, magenta; cPGAP1$^{H443N}$, light pink). The 1-alkyl is shown in ball-and-stick representation. PLM (green) and cholesterol (CHL, yellow) are from cPGAP1$^{S327A}$. cPGAP1 is shown as a surface representation with the transmembrane (TM) domain in blue and the lipase domain in orange. **e** Effect of CHL (triangle, pink) and cholesteryl hemisuccinate (CHS, circle, blue) on cPGAP1 activity. Average ± s.d. from three independent experiments are plotted. Statistical analyses were performed using one-way ANOVA followed by Sidak's multiple comparisons test. Source data of (**b**, **e**) are provided as a Source Data file. FSEC, fluorescence-detection size exclusion chromagraphy; GPI-AP, glycosylphosphatidylinositol-anchored protein.

suggests the possibility that the substrate initially anchors to the enzyme by inserting the 2-acyl into the side pocket (Fig. 4d), resembling a compass needle (Fig. 5d). Subsequently, the 1-alkyl chain, analogous to a compass pencil, swings towards the enzyme (Fig. 5d). During product release, GPI-AP$_2$ may undergo a similar motion but in the reverse direction (Fig. 5d), exposing the 1-alkyl chain to the bulk membrane. This exposure may facilitate downstream enzymes like PGAP5 to bind the alkyl chain (along with the exposed glycan), potentially aiding in substrate channeling.

Given that PLM remains bound to cPGAP1 during purification and is enclosed by GPI-AP$_2$ in the guitar-shaped cavity (Fig. 4b), it is likely that the fatty acid is released after GPI-AP$_2$.

In cPGAP1$^{S327A}$, a blob of density fitting well for a cholesterol molecule (Fig. 2f) occupies the 1-alkyl position of the TGP$_2$ in cPGAP1$^{H443N}$ (Fig. 5d). To explore possible functional consequences of this observation, we conducted enzymatic assays under varying cholesterol concentrations while maintaining a constant detergent concentration. Cholesterol had no discernible effect on cPGAP1 activity at

0.08 mol% or lower concentrations. However, inhibition became evident at 0.33 mol%, reducing cPGAP1 activity by approximately half at 5 mol% (Fig. 5e), a concentration (in relation to total lipids) found in the ER membranes of mammalian cells[60]. This effect appeared somewhat specific, as the hemisuccinate derivative cholesteryl hemisuccinate (CHS) showed lesser inhibition (Fig. 5e). These results suggest the functional significance of the binding between the 1-alkyl moiety and enzyme, and a regulatory role of cholesterol in substrate binding, assuming a causative relationship between cholesterol binding at this site and the observed inhibition.

## Discussion

The post-attachment GPI remodeling pathway plays a crucial role in the maturation, localization, and biological function of GPI-APs for all eukaryotic cells[2]. PGAP1 is the first GPI-AP remodelase of this pathway, and its action initiates quality control and ER export of GPI-APs. In this study, we investigated cPGAP1, a cross-species active GPI-AP remodelase in yeast and humans, using biological, biochemical, and structural

approaches. Our findings uncover mechanisms for substrate recognition and catalysis, providing a reliable template for homology modeling of orthologs. This knowledge can be harnessed for designing inhibitors as research tools or antibiotics against pathogens.

The study of GPI-AP biogenesis enzymes has been challenging due to the lack of commercial substrates and convenient assay methods. However, the engineered GPI-AP substrate and the sensitive, quantitative, and convenient fluorescence assay developed in this study allowed for the kinetic characterization of PGAP1. These strategies can be adapted to study other GPI-AP enzymes and may inspire new approaches to investigate other lipid modification processes such as acylation, prenylation, phospholipid modification, and cholesterylation.

As an Asp-His-Ser triad lipase, PGAP1 exhibits atypical features. Notably, all the triad residues possess variable tolerance to mutations. The non-essentiality of Asp407 suggests His443 has a relatively high $pK_a$. While His443 is catalytically more important than Asp407, it is still not essential, indicating that the microenvironment provides some level of Ser327 activation. Typically, the alanine mutation of the serine abolishes the triad function, but rare exceptions have been documented[61,62]. In the case of cPGAP1 S327A, nearby solvent water may occupy the serine hydroxyl position[59] and be activated by the histidine to perform the hydrogen bond relay that is usually carried out by the serine.

Another complexity associated with cPGAP1 S327A is its environment-dependent leaky activity. While it shows no activity in the cell-based FACS assay with co-expressed substrates and in detergents with separately added substrates, deacylated products were found in the membranes of broken cells co-expressing TGP$_3$. This suggests that cPGAP1 S327A may adopt slightly different conformations between live and dead cell membranes with distinctive functional consequences.

Our results provide a cautionary note for interpreting dead mutations of PGAP1 or other serine hydrolases. For instance, a previous study reported that the mouse PGAP1 S174A mutant (the cPGAP1 S327A counterpart) could modify Wnt[40]. Since the modification was sensitive to a GPI-specific phospholipase, it was interpreted as a GPI-like modification but catalyzed by regions outside the lipase domain. Based on our findings, however, it is also possible that the GPI-like modification of Wnt was indeed GPI-anchoring, and the impaired triad catalyzed the deacylation reaction.

Efficient inositol deacylation depends on the calnexin cycle through a mechanism in which calnexin increases the ER retention time of GPI-APs for the PGAP1 reaction[18,19]. From a biochemical standpoint, this implies slow PGAP1 kinetics, which is consistent with our enzymatic data, as cPGAP1 exhibits a $K_{cat}$ of only 7 min$^{-1}$. The slow activity may have evolved for quality control purposes to ensure enough time for GPI-AP sorting. Therefore, PGAP1 may act as a rate-limiting remodelase, thus serving as a checkpoint for quality control of GPI-APs. Alternatively, the low apparent activity may be linked to the species-specific glycan preferences. Specifically, the Man4 modification is required for GPI-AP biogenesis in *S. cerevisiae*, and by reasonable extension, in *C. thermophilum* which is also a fungus. Although the involvement of the PGAP1 step in the Man4 requirement is unknown, the product-bound structures of TGP$_3$ trapped in our study contained the Man4 moiety, despite its scarcity in mammalian cells[63]. In addition, the purification yield of cPGAP1 H443N/S327A-TGP$_3$ complex was approximately 10% at the second affinity chromatography step, implying a heterogeneous nature of TGP$_3$ with varying suitability as cPGAP1 substrates. Supporting this speculation, our prolonged enzymatic assays using saturating wild-type enzyme only achieved ~90% completion. Moreover, Man4 formed three H-bonds with cPGAP1 (Fig. 4e), with noteworthy observations that related mutations increased activity (Table S2). Future experiments using cell lines overexpressing PIGZ (the enzyme responsible for Man4 modification)[2]

or PIGZ-KO cells may help clarify whether Man4-modification increases cPGAP1 activity. A yet another hypothesis is that PGAP1 might exhibit higher activity in native membranes, a possibility requiring a new assay format to test.

Finally, the function of the jelly-roll domain is worth discussing. Although the deletion leads to instability, its physiological function likely extends beyond stabilization. Jelly-rolls are involved in various protein-protein (viral capsid)[64], protein-carbohydrate[65], and protein-lipid[66] interactions. Intriguingly, p24 proteins[67], including cargo adaptors for GPI-APs, and several other proteins involved in the Golgi dynamics and secretion, contain a jelly-roll Golgi dynamics (GOLD) domain[68,69]. Given the functional connection between PGAP1 and the secretion machinery, it is possible that its jelly-roll domain serves as an adaptor to interact with some of these GOLD proteins for the relay and transport of GPI-APs.

## Methods

### Molecular cloning

All PGAP1 constructs used in this study contain the full-length protein unless otherwise specified, such as the truncation constructs for the jelly-roll domains.

The DNA fragment encoding the *Chaetomium thermophilum* PGAP1 (GenBank ID XP_006693856.1 [https://www.ncbi.nlm.nih.gov/protein/XP_006693856.1], cPGAP1 hereafter) with flanking 5' *BamH*I and 3' *Kpn*I sites was synthesized by Azenta. The fragment was digested with the two restriction enzymes and T4-ligated into pBTSG (Addgene 159420)[70] pretreated with the DNA enzymes. This construct expresses a protein, from N- to C-terminus, in the following order: cPGAP1, 3 C protease site, a thermostable green fluorescence protein (TGP)[70], a twin Strep-tag, and a His-tag. The TGP-containing construct (pBTSG-cPGAP1, Fig. S11a) was used for the structure determination of cPGAP1 wildtype in nanodisc (cPGAP1$^{apo}$). A similar construct was made for the human hPGAP1 (pBTSG-hPGAP1) (GenBank ID NP_079265.2 [https://www.ncbi.nlm.nih.gov/protein/NP_079265.2]). An additional construct (pBTSG2-hPGAP1/cPGAP1) was made by replacing the cytomegalovirus promoter with a TATA-box-containing promoter (used in Fig. 1b).

For test-tube enzymatic studies where GPI-anchored TGP was used as the substrate, the TGP-tag on cPGAP1 was exchanged into Strep-tagged mCherry (pBCh-cPGAP1, Fig. S11a, ii) to avoid interferences for the detection of the TGP substrate/product. The mCherry tag has the sequence of Uniprot ID A0A366VY15 [https://www.uniprot.org/uniprotkb/A0A366VY15/entry] with the following four mutations that extends its fluorescence lifetime: W148S, I166V, Q168Y and I202R (W143S, I161V, Q163Y, and I197R, in ref. 71). Mutants for biological and biochemical assays were generated based on pBCh-cPGAP1 by site direct mutagenesis.

For the purification and structure determination of the enzyme-product complex (cPGAP1$^{H443N}$ and cPGAP1$^{S327A}$), His-tagged pBCh-cPGAP1 (Fig. S11a, ii) was used.

Deletion constructs for the jelly-roll subdomains were made by deletion PCR. In hPGAP1Δ jelly-roll, residues Lys331-Thr581 (inclusive) were replaced with a 12 Gly-Ser linker. In cPGAP1 Δ jelly-roll 1 (the upper subdomain in Fig. 3a), cPGAP1 Δ jelly-roll 2 (the bottom subdomain in Fig. 3a), and cPGAP1 Δ jelly-roll, the regions replaced by a 12-Gly-Ser linker include residues Ala498-Ser673, His680-Thr802, and Glu495-Val803, respectively.

To produce GPI-anchored TGP in HEK293T cells (Cat. CRL-3216, ATCC, not authenticated), a chimera TGP-DAF construct (pCD55$^{chimera}$) was made (Fig. S4a). The construct was based on the pBTSG backbone and contained DNA fragment encoding the following elements with the order from N- to C-terminus: the N-terminal signal peptide region of DAF (residues Met1 to Gly34, Uniprot ID P08174 [https://www.uniprot.org/uniprotkb/P08174/entry]), TGP, a Strep II tag, and the ω-minus region plus the C-terminal signal peptide of DAF (residues

Pro345 to Thr381). The construct was expected to produce Strep-tagged TGP$_2$ in wildtype HEK293T cells and TGP$_3$ in PGAP1-KO cells. A similar construct was made by replacing the Strep II tag with a Flag tag. The Strep-tagged construct was used for biochemical and structural studies, while the Flag-tagged construct was used for flow cytometry analysis. For simplicity, both constructs are referred to as TGP$_3$/TGP$_2$ in the main text.

To produce PI-PLC, a DNA fragment encoding residues Ala32 to Glu329 (without signal peptide) of the *Bacillus cereus* PI-PLC (Uniprot ID P14262 [https://www.uniprot.org/uniprotkb/P14262/entry])) was synthesized with *Bsp*QI site at the N- and C-terminus (Azenta). The fragment digested with *Bsp*QI (Cat. R0712L, New England Biolabs) was T4-ligated into *Bsp*QI-treated pSb-init[72] which contains an N-terminal signal peptide for periplasmic expression in *Escherichia coli*.

The constructs for yeast genetics are described latter with the genome integration protocols.

All constructs in this study were verified by Sanger sequencing.

### Generation of PGAP1-KO HEK293T cells

Two pairs of sgRNA oligos: Ex9_Fwd (5′- CACCGTTCTAGTAAAAGTG TCCAAA-3′) and Ex9_Rev (5′- AAACTTTGGACACTTTTACTAGAAC-3′); and Ex10_Fwd (5′- CACCGATTTTTCTATGATTTTCAAG-3′) and Ex10_Rev (5′- AAACCTTGAAAATCATAGAAAAATC-3′) were designed using the online server (http://cistrome.org/SSC/)[73] to target the exon 9 and 10 of the *PGAP1* gene in HEK293 cells. The construct was verified by Sanger sequencing. The oligos were designed to have sticky ends of Type IIs restriction enzyme *Bbs*I. The oligos were dissolved in a buffer containing 0.2 M NaCl, 0.1 mM EDTA, and 10 mM Tris HCl pH 7.5 and mixed to have 10 μM of each in a PCR tube. After heating at 95 °C for 3 min, the oligo pairs were annealed by gradual cooling from 94 °C to 25 °C at 1-°C gradients with an 11-s incubation time for each step. The annealed mix (1 μL) was ligated into pX330 (50 ng) predigested with *Bbs*I (Cat. R3539S, New England Biolabs) using T4 ligase (Cat. EL0014, Thermo Fisher Scientific). The ligation products were transformed into *E. coli* DH5α cells. Positive colonies were identified by Sanger sequencing.

For CRISPR-Cas9 gene editing, 8 μg of the sgRNA-carrying plasmids, 0.16 μg of pMaxGFP, 16 μL of P3000 (Cat. L3000008, Thermo Fisher Scientific), and 250 μL of Opti-MEM medium (Cat. 31985070, Thermo Fisher Scientific) were first mixed before being added to another premix containing 16 μL of Lipofectamine 3000 and 250 μL of Opti-MEM medium at room temperature (RT, 20–22 °C) for 15 min. After incubation, the mixture was added dropwise to a 6-cm dish containing HEK293T cells with 70%–90% confluency. Cells were cultured in Dulbecco's Modified Eagle Medium (DMEM, Cat. C11995500BT, Thermo Fisher Scientific) supplemented with 10% fetal bovine serum (FBS, Cat. 40130ES76, Yeasen, Shanghai, China) in an incubator with 5% CO$_2$ at 37 °C. After 24 h, cells were washed with 2 mL of phosphate-buffered saline (PBS), digested with 0.5 mL of 0.1% trypsin (Cat. 25200056, Thermo Fisher Scientific) at 37 °C for 3 min, and re-suspended in 3 mL of DMEM and 10% FBS. Cells were collected by centrifugation at RT at 300 g for 5 min, washed with 10 mL of PBS, and re-suspended with 0.5 mL PBS. The resuspension was applied to a BD FACSAria Fusion machine operated under software BD FACS Diva (version 8.0.3) for FACS. The top 5% green fluorescence protein (GFP)-positive cells (~40,000) were collected and serially diluted using DMEM supplemented with 10% FBS before being seeded into 96-well plates. Wells containing single colonies were noted. After 10–12 d in a stationary CO$_2$ incubator, cells from single colonies were detached using trypsin, resuspended in 200 μL of DMEM and 10% FBS, and divided into two aliquots (160 μL and 70 μL), one for scaling up and the other for PCR-identification. Positive clones were expected to produce a DNA fragment of approximately 1 kb while that for the wildtype cells was expected to run at 2.5 kb on agarose gel after PCR reaction using the primer pair 5′- ATTTCCCTATGATGATGCTGGT -3′ and 5′-

GCCAATGGAAAACAAAATTCCCTT -3′. Genomic deletions in positive clones were verified by Sanger DNA sequencing.

### Protein expression and purification – wildtype cPGAP1 for structural study

The day before transfection, 1 L of Expi293 (Cat. A14527, ThermoFisher Scientific, not authenticated) cells were diluted to a density of $1 \times 10^6$ mL$^{-1}$ and cultured at 37 °C in a CO$_2$ shaking incubator. A total of 2 mg plasmid (Fig. S11a) and 4 mg polyethylenimine (PEI, Cat. 23966-1, Polysciences) were mixed with 50 mL of medium in two separate tubes for 3 min before being pooled together for incubation at RT for 20 min. The mixture was then added into 1 L of cell culture which typically had a density of $2 \times 10^6$ mL$^{-1}$. In addition, sodium valproate (Cat. P4543, Sigma) was supplemented at a final concentration of 2 mM to improve protein expression. Cells were harvested after 48 h, washed with PBS buffer, snap-frozen with liquid nitrogen, and stored at -80 °C before use.

All the purification procedures were conducted at 4 °C. Cells from 0.3 L of culture were resuspended in a Buffer A (150 mM NaCl, 50 mM HEPES pH 7.5) supplemented with 1 mM EDTA, 2 mg mL$^{-1}$ iodoaceta-mide (IAM), 1 mM phenylmethylsulfonyl fluoride (PMSF), and 1 ×pro-tease inhibitor (Cat. B14001, Biomake). Cells were lysed by probe sonication for 10 min with 30% power and 3s-on/5s-off cycles (SCI-ENTZ). The lysate was clarified by centrifugation at 3000 g for 10 min. The supernatant was centrifuged at 48,000 g for 1 h. The membrane pellets were then resuspended in Buffer A supplemented with 1 mM PMSF, 1 × cocktail, 2 mg mL$^{-1}$ IAM before being solubilized using 1%(w/v) dodecylmaltoside (DDM, Cat. DDM990, Bluepus, Shanghai, China), 0.2%(w/v) cholesteryl hemisuccinate (CHS, Cat. CHS990, Bluepus) for 2 h. Cell debris was removed by centrifuging at 48,000 g for 1 h. The supernatant containing solubilized cPGAP1 was collected and mixed with 2 mL of Strep Tactin beads (Cat. SA053100, Smart-lifesciences) pre-equilibrated with 0.03% DDM/CHS in Buffer B (150 mM NaCl, 20 mM HEPES pH 7.5) and stirred gently for 2 h. The mixture was then pooled into a gravity column for purification and detergent exchange. The beads were washed successively with 5 column volume (CV) of 0.03% DDM/CHS and 0.008% glyco-diosgenin (GDN, Cat. GDN101, Anatrace), 2.5 CV of 0.1% GDN, 2.5 CV of 0.016%(w/v) GDN, and 5 CV of 0.008% GDN in Buffer B. The beads were incubated for 15 min after the second wash step to facilitate detergent exchange. cPGAP1 was eluted with 5 mM D-desthiobiotin (Cat. Sc-294239A, Santa Cruz), 0.008% GDN in Buffer B. The elution was used for nanodisc reconstitution immediately.

### Nanodisc reconstitution

The membrane scaffolding protein (MSP) 1E3 was expressed and purified[74] as follows. *E. coli* BL21 (DE3) cells expressing a plasmid encoding MSP1E3 were resuspended in the Lysis Buffer (1 %(w/v) Triton X-100, 10 mg L$^{-1}$ DNase, 1 mM PMSF, 8 mM MgCl$_2$, 150 mM NaCl, and 10 mM sodium phosphate pH 7.2) and lysed by sonication. The cell lysate was adjusted to contain 10 mM imidazole, clarified by cen-trifugation, and loaded onto a Ni-NTA column. The column was washed successively with 5 CV of 10 mM imidazole and 1%(w/v) Triton X-100, 10 CV of 20 mM imidazole and 50 mM sodium cholate, and 15 CV of 40 mM imidazole in 300 mM NaCl, 40 mM Tris-HCl pH 8.0. MSP1E3 was eluted using 300 mM imidazole in 300 mM NaCl, 40 mM Tris-HCl pH 8.0. Protein was quantified by absorbance at 280 nm measured on a Nanodrop machine with the theoretical molar extinc-tion coefficient of 32,430 M$^{-1}$ cm$^{-1}$.

To prepare the lipid stock, 9.8 mg of POPS and 2.5 mg of GDN were co-dissolved in chloroform/methanol (1: 1, vol: vol). The solution was dried under argon, lyophilized, and dissolved in 0.92 mL of Buffer B. The wildtype cPGAP1 in GDN purified above was incubated with the MSP1E3 and 1-palmitoyl-2-oleioyl-phosphatidylserine (POPS) at 1: 4: 200 molar ratio with a total of volume of 2 mL. After a 2-h incubation,

Biobeads (Cat. 152-3920, Bio-Rad) were included to a final concentration of 200 mg mL$^{-1}$ in two steps with a 2-h incubation after the first addition and a 16-h incubation after the second addition. The Biobeads were then removed from the reconstitution mixture by centrifugation, and the supernatant was concentrated to 3 mg mL$^{-1}$ using a 100-kDa cutoff membrane concentrator (Cat. UFC810096, Merck millipore) before being applied onto a Superose 6 increase 10/300 GL column (Cat. 29-0915-96, Cytiva) with Buffer B as the running buffer. Peak fractions were pooled and concentrated to 15 mg mL$^{-1}$ of total protein for cryo-EM sample preparation. Protein concentration was quantified by absorbance at 280 nm measured using a Nanodrop machine with the theoretical molar extinction coefficient of 256,765 M$^{-1}$ cm$^{-1}$ assuming a molar ratio of 1: 2 (cPGAP1: MSP1E3). Purified protein was analyzed by SDS-PAGE and the bands were visualized using a portable TGreen Transilluminator (OSE-470, Tiangen, Beijing, China) for in-gel fluorescence. Gel images were taken using a smartphone.

### Protein expression and purification – cPGAP1 H443N/cPGAP1 S327A in complex with GPI-anchored TGP for structural studies

PGAP1-KO HEK293T cells were adapted to suspension culture for the expression of the complexes. To adapt the adherent cell line for suspension, cells were cultured in 30 mL sera-free medium (Cat. 1000, Union, Shanghai, China) for two days at 37 °C in a CO$_2$ (5%) incubator shaking at 125 r.p.m.. Adapted cells were co-transfected with the construct expressing Strep-tagged CD55-TGP (TGP$_3$) (Fig. S4a) and His-tagged pBCh-cPGAP1 S327A or pBCh-cPGAP1 H443N (Fig. S11a, ii) using PEI-mediated method as outline above. The plasmid ratio was 3:1 (wt: wt, cPGAP1 mutant: pTGP$_3$). All purification steps were carried out at 4 °C. Cell pellets from 5 L of cell culture were resuspended with Buffer C (150 mM NaCl, 50 mM Tris pH 8.0) supplemented with protease inhibitor cocktail. Cells were lysed by probe sonication. Cell membranes were harvested by centrifugation at 48,000 g for 1 h, solubilized with 1%(w/v) DDM, in Buffer A supplemented with protease inhibitor cocktail for 1.5 h. The supernatant was collected and mixed with 6 mL pre-equilibrated Strep Tactin beads with gentle stirring for 1.5 h. The mixture was loaded into a gravity column, and the beads were sequentially washed with 5 CV of 0.03% DDM, 0.01% GDN, 1.5 CV 0.1% GDN in Buffer B. After a 15-min incubation, the beads were washed again with 1.5 CV 0.1% GDN in Buffer B before being eluted with 5 mM D-desthiobiotin and 0.008% GDN in Buffer B. A second affinity chromatography was performed to remove free GPI-anchored TGP (TGP$_3$/TGP$_2$). To this end, pooled fractions were incubated with 3 mL pre-equilibrated Ni-NTA resins supplemented with 10 mM imidazole with mild agitation for 1.5 h. The beads were packed into a gravity column, washed with 3 CV 10 mM imidazole and 0.008% GDN in Buffer B, and eluted with 0.3 M imidazole, 0.008% GDN in Buffer B. Pooled fractions were concentrated using a 100-kDa cut-off concentrator before being further fractioned by gel filtration in a RioRad NGC machine controlled by ChromLab (version 5.0.2.11) using a Superose 6 increase 10/300 GL column pre-equilibrated in 0.008% GDN in Buffer B. Peak fractions were pooled and concentrated to 10 mg mL$^{-1}$ (cPGPA1 H443N) or 14.9 mg mL$^{-1}$ (cPGAP1 S327A) for cryo-EM grid preparation. Protein concentration was determined by measuring A$_{280}$ using a theoretical molar extinction coefficient of 222,337 M$^{-1}$ cm$^{-1}$ assuming an equimolar stoichiometry between cPGAP1 (mutants) and TGP$_2$ (the product).

The purification of a substrate-bound complex using the double mutant cPGAP1 S327A/H443N was unsuccessful owing to low binding activity between the co-expressed cPGAP1 mutant and TGP$_3$.

### Protein expression and purification – cPGAP1-mCherry and mutants for biochemical assay

For biochemical assays using TGP$_3$ as the substrate, cPGAP1 was fused with mCherry rather than TGP to avoid background issues (Fig. S11a). The expression and purification procedures were similar to cPGAP1-TGP fusion proteins with slight differences. All purification procedures were performed at 4 °C. Typically, cells from 10 mL of culture were harvested and resuspended in Buffer A with 3 mL buffer for every gram of wet cell mass. Cells were directly lysed by incubation with 1% DDM for 1 h. Cell debris was removed by centrifugation at 21,000 g for 1 h, and the supernatant fraction was incubated with 10-30 uL pre-equilibrated Strep Tactin beads for 1 h in a low-binding tube (Cat. MCT-150-L-C, Axygen). The mixture was centrifuged and the supernatant was removed using a micropipette. The beads were washed four times using 7 bed volume (BV) of 0.03% DDM in Buffer B in each wash. The washing fraction was removed by centrifugation, and cPGAP1 mutants were obtained by two successive elution steps, each with 2−3 BV of 5 mM D-desthiobiotin and 0.03% DDM in Buffer B. The concentration was determined using calibrated mCherry fluorescence (60 RFU for 1 mg L$^{-1}$) measured using 0.1 mL sample in a half-area 96-well plate at the excitation/emission wavelength pair of 587/615 nm in a plate reader (SpectraMax M2, Molecular Devices). Protein samples were divided into 5-µL aliquots, flash frozen in liquid nitrogen, and stored at −80 °C before use.

The integrity of the cPGAP1 mutants was assessed by in-gel fluorescence using a Typhoon scanner (FUJI, Starion FLA-9000). All mutants except V163F showed negligible degradation (Fig. S8).

### Protein expression and purification - TGP$_3$ and TGP$_2$

To preserve the inositol acyl of the GPI-anchored TGP$_3$, PGAP1-KO HEK293T cells were used for expression. Cells adapted as mentioned above were transfected with the pCD55$^{chimera}$ plasmid. Cell pellets were resuspended in Buffer A (3 mL for each gram of wet cell mass) supplemented with 1 mM PMSF and 1 × cocktail, sonicated as aforementioned, and centrifuged at 48,000 g for 1 h. The membranes were resuspended, washed once in high salt buffer (1 M NaCl, 10 mM HEPES pH 7.5), and solubilized for 1 h using 1% DDM in Buffer B. The mixture was heated at 75 °C for 5 min and clarified by centrifugation 20,000 g for 30 min. The supernatant containing the heat-resistant TGP$_3$ was then incubated with the pre-equilibrated Strep Tactin beads at 4 °C for 2 h. The beads were packed into a gravity column, washed by 15 CV of 0.03% DDM in Buffer B, and eluted with 5 mM D-desthiobiotin and 0.03% DDM in Buffer B. Pooled fractions were further fractioned by gel filtration using a Superdex 200 increase 10/300 GL column (Cat. 28-9909-44, Cytiva) with 0.03% DDM in Buffer B as the running buffer. Fractions were concentrated to 52 mg mL$^{-1}$ using a 30-kDa cut-off concentrator, divided into 5-µL aliquots, flash frozen and stored at −80 °C before use. The concentration was determined using calibrated TGP fluorescence (606 RFU for 1 mg L$^{-1}$) measured using 0.2 mL sample in a 96-well plate at the excitation/emission wavelength pair of 488/512 nm in a plate reader (SpectraMax M2, Molecular Devices).

The purification of TGP$_2$ was performed using the same procedure except that the protein was expressed in wildtype Expi293 cells.

### Protein expression and purification – PI-PLC

*Escherichia coli* MC1061 cells carrying the pSb-int-PI-PLC plasmid were cultured in 110 mL Terrific Broth (TB) (2.4%(w/v) yeast extract, 1.2%(w/v) tryptone, 0.4%(w/v) glycerol) supplemented with 25 µg mL$^{-1}$ chloramphenicol at 37 °C in an orbital shaker with 220 rpm for overnight. Cells were seeded into 1 L of the same medium at a 1:100 dilution and cultured at 37 °C for 2 h, after which the temperature was reduced to 22 °C. After another 2 h, cells were induced with 0.2% arabinose for 16 h before being harvested by centrifugation. Cell pellets from 1 L of culture were resuspended in 20 mL of TES buffer (200 mM HEPES pH 7.5, 0.5 mM EDTA, 500 mM sucrose). Periplasmic extraction and purification were performed at 4 °C as follows. Cells were dehydrated in the TES buffer for 30 min before rapidly re-hydrated by the addition of 40 mL pre-cooled water into the mixture. Periplasmic extraction was recovered by centrifugation at 20,000 g for 30 min. The supernatant was added with 150 mM NaCl, 2 mM MgCl$_2$, 20 mM imidazole before

being incubated with pre-equilibrate 0.5 mL Ni-NTA (Cat. 1018401, Qiagen). After 2 h of batch binding, the beads were packed into a gravity column, washed with 20 CV 30 mM imidazole in Buffer B. PI-PLC was eluted by 5 CV of 300 mM imidazole in Buffer B, snap-frozen in liquid nitrogen, and stored at −80 °C before use. PI-PLC was quantified using molar extinction coefficient of 66,810 $M^{-1}$ $cm^{-1}$ with absorbance at 280 nm measured in a Nanodrop spectrophotometer.

## Flow cytometry

Wide type, PIGK-KO[46] (negative control for background staining), and human PGAP1-KO HEK293T cells were cultured in Dulbecco's Modified Eagle Medium (DMEM) supplemented with 10% FBS and 100 units mL$^{-1}$ of penicillin and 100 μg/mL streptomycin in a 24 well plate one day before transfection. The transfection was performed as follows: 0.5 μg total plasmid was mixed with 1 μL P3000 (Cat. L3000008, Thermo Fisher Scientific) in 25 μL opt-MEM medium and added to another premix of 1 μL Lipofectamine 3000 in 25 μL opt-MEM medium at RT for 15 min. The mixture was then added dropwise into the cell culture in a 24 well plate. For Fig. 1b, a total of 1 μg plasmid was used for transfection as above. After another culturing at 37 °C for 48–72 h, cells were collected and washed once using PBS, resuspended in 500 μL PBS and digested with or without10 ug of PI-PLC at 37 °C for 1.5 h. Cells were washed again using PBS, followed by antibody staining.

For experiments using CD59 as the GPI-AP reporter, phycoerythrin (PE)-labeled anti-CD59 antibody (Cat. 12-0596-42, Clone OV9A2, Thermo Fisher Scientific) was used as a 500-fold dilution for incubation with the cells for 15 min at dark. Cells were washed once and resuspended in 0.3 mL of PBS. Cells were analyzed using flow cytometry (Beckman CytoFlex LX) using two wavelength-pairs (488/525 nm for TGP-tagged PGAP1, 561/585 nm for PE).

For experiments using Flag-tagged TGP$_2$ or TGP$_3$ as the GPI-AP reporter, the procedure was the same as above, except that a ratio of 3:1 (wt:wt, enzyme:substrate) was used and the staining was performed with different antibodies as below. Thus, Alexa647-labeled anti-Flag antibody (Cat.15009 S, Clone D6W5B, Cell Signaling Technology) was used at a 100-fold dilution for incubation. Cells were analyzed using flow cytometry (Beckman CytoFlex LX machine controlled by software CytExpert 2.4.0.28.) using three wavelength pairs (488/525 nm for the expression of the TGP-tagged substrate, 561/610 nm for mCherry-tagged PGAP1, and 638/660 nm for surface display of TGP$_2$/TGP$_3$). The FACS data were analyzed using FlowJo v10.8.1.

FACS experiments were carried out following the strategies in Fig. S12. In all FACS runs, cells were first gated to select living cells and single cells. For cell surface staining of CD59, the expression of TGP-fused hPGAP1/cPGAP1/mutants was gated by the fluorescence of TGP (488/525 nm). This population was further analyzed for phycoerythrin (PE)-positivity (561/585 nm) as an indication of the cell surface staining of CD59 via its antibody (Figs. 1b, 2e, 3f).

For cell surface staining of Flag-tagged TGP$_3$/TGP$_2$, cells expressing Flag-tagged TGP$_3$/TGP$_2$/TGP$_0$ were first gated by TGP fluorescence to eliminate non-expressing cells (total TGP expression). The TGP-positive cells were further analyzed by allophycocyanin (APC) fluorescence (638/660 nm from anti-Flag antibodies), which indicates cell surface expression of Flag-tagged TGP$_3$/TGP$_2$ (Fig. S4b). In cells co-expressing mCherry-tagged cPGAP1/mutants, the TGP positive cells were gated by mCherry fluorescence (for cPGAP1 expression) before being gated by APC fluorescence (from anti-Flag antibodies) for the surface expression of TGP$_3$/TGP$_2$ (Fig. 2g). All the commercial antibodies used in this study were validated by the manufacturers (see Report Summary for details).

## FSEC-based activity assay

For the biochemical assay of cPGAP1 using separately purified TGP$_3$ (substrate), the following procedure (Fig. S3, Type 3) was used.

To minimize the use of the TGP$_3$ substrate, enzymatic reactions were carried out in a 0.2 mL PCR-tube with a total volume of 2 μL. The system was found to be reproducible. The reaction mixture contained 1 μL of TGP$_3$ (substrate), and 1 μL of cPGAP1 (wildtype and mutants), purified in DDM. When dilutions are required, 0.03% DDM in Buffer B was used. The reaction was carried out in a PCR machine at 20 °C. The reaction time was typically 20 min. When longer time was used, the conditions were determined to be in the linear region regarding the reaction time and activity under given conditions such as enzyme concentrations. After the PGAP1 reaction, the mix was added with 18 μL of 0.03% DDM, 0.006% CHS in Buffer B and heated at 75 °C for 5 min to inactivate PGAP1 (the inactivating condition was pre-determined). The mix was cooled to 4 °C and either directly digested with PI-PLC, or diluted by up to 100 folds depending the TGP$_3$ concentrations before PI-PLC digestion. The amount of PI-PLC (6.5 μg per 100 micrograms of TGP$_3$) was pre-determined to be at 10 folds excess of a saturation level. The PI-PLC digestion was performed at 37 °C for 10−30 min. The mixture was centrifuged at 21,000 g for 10 min at 4 °C before being applied onto a Sepax Zenix-C SEC-300 column connected to a high-performance liquid chromatography (HPLC) system equipped with a fluorescence detector (RF-20A, Shimadzu). The fluorescence-detection size exclusion chromatography (FSEC) profile was monitored by fluorescence at the excitation/emission pair of 482/508 nm using software LabSolutions (version 5.87). The TGP$_0$ (product) was quantified by calibrated FSEC peak intensity obtained using known amounts of TGP$_0$.

For the Michaelis-Menten kinetics of the wildtype cPGAP1, the enzyme concentration was 8 μg mL$^{-1}$. The TGP$_3$ concentration was varied from 2.5 to 800 μM (0.065−20.8 mg mL$^{-1}$). Kinetic data were fitted using GrapPad Prism 8.

For the assay of mutations, the reaction contained 100 μM (2.6 mg mL$^{-1}$) of TGP$_3$. An initial assay was performed using a blanking concentration of enzyme (cPGAP1 mutants) at 32 μg mL$^{-1}$. The enzyme concentration and/or the reaction time were then reduced (for the gain-of-function mutants) to have a similar reaction rate to the wildtype, or increased (for the loss-of-function mutants) to ensure enough signal.

For the assay of cPGAP1 S327A with separately purified TGP$_3$ under prolonged incubation conditions, S327A and TGP$_3$ were mixed at equimolar concentrations (0.244 mg mL$^{-1}$ for S327A and 0.04 mg mL$^{-1}$ for TGP$_3$) and incubated for 46 h at 4 °C. PI-PLC treatment and FSEC were performed as above.

For the assay of the cPGAP1 S327A mutant in the membranes using co-expressed TGP$_3$ as the substrate (Fig. S3, Type 2), cells from 150 mL of culture were lysed by sonication. Membranes were harvested by centrifugation at 48,000 g for 1 h at 4 °C. The membranes were washed once with high-salt buffer (1 M NaCl, 10 mM HEPES pH7.5), resuspended in Buffer B, heated at 75 °C for 5 min, and treated with or without 7 μg PI-PLC for 30 min at 37 °C. The reaction mix was clarified by centrifugation at 21,000 g for 30 min. The supernatant was directly analyzed using FSEC. The double mutant cPGAP1 S327A/H443N was used as a negative control.

For the assay of TGP$_2$/TGP$_3$ composition in the cryo-EM samples of cPGAP1$^{S327A}$ and cPGAP1$^{H443N}$, the protein specimen used for grid preparation was diluted to 38 μg mL$^{-1}$ (cPGAP1$^{H443N}$) or 75 μg mL$^{-1}$ (cPGAP1$^{S327A}$) and treated with PI-PLC. The reaction mixture was then analyzed by FSEC as outlined above.

To investigate the effect of cholesterol on cPGAP1 activity, the assay was performed in 5%(w/t) DDM, 150 mM NaCl, 20 mM HEPES pH 7.5 with or without cholesterol/cholesteryl hemisuccinate (CHS) at concentrations ranging from 0.002 to 5 mol% (in relation to the molarity of the DDM detergent). Purified cPGAP1-mCherry was incubated with the assay mix for 30 min before purified TGP$_3$ was added to the mix. The final concentrations for the enzyme and substrate were at 32 μg mL$^{-1}$ and 0.1 mM, respectively. The reaction (2 μL) was proceeded

for 40 min at 20 °C in a PCR machine as mentioned above. PI-PLC treatment and FSEC analysis were performed as described above. Statistic analyses (see figure legends for details) were performed using GrapPad Prism 8.

## Triton X-114 phase separation

Triton X-114 (Cat. X114, Sigma) was condensed by three times of phase separation as follows. The solution was added to ice-cold PBS and incubated at 37 °C for 15 min. The aqueous phase was separated from the viscous detergent phase by centrifugation at 1000 g for 30 min at 30 °C. To prepare working solutions, condensed Triton X-114 was diluted using Buffer B, and the concentration was determined using an extinction coefficient[75] of 206.6 $(mg/mL)^{-1} cm^{-1}$ with absorbance at 280 nm measured in a Nanodrop spectrophotometer.

A reaction mixture (Fig. S3, Type 4) was made by mixing 0.2 µg of $TGP_3$ (substrate), 1.2 µg of cPGAP1 (enzyme) in 0.03% DDM in Buffer B. After 10 min at 20 °C, PI-PLC was added to a final concentration of 200 µg mL$^{-1}$. After 10 min at 20 °C, the mixture was adjusted to contain 2% of Triton X-114, vortexed for 1 min, and placed on ice-water for 10 min to form a homogenous solution. The mixture was then incubated at 37 °C for 15 min, and phase separation was performed as described above. The aqueous phase was extracted with an equal volume of 2% Triton X-114, while the detergent phase was extracted with an equal volume of Buffer B. Phase separation was repeated. The aqueous phase and detergent phase from the second step were analyzed by SDS-PAGE and the TGP bands were visualized by in-gel fluorescence (Typhoon FLA-9000 with Image Reader Ver.1.0). Reactions omitting PGAP1 or PI-PLC were performed as controls.

## Thin layer chromatography

To confirm the production of the fatty acid product, thin-layer chromatography (TLC) was carried out as follows. A reaction mix including 0.5 mg of $TGP_3$ and 24 µg of cPGAP1 was incubated at 26 °C 1 h. The reaction was extracted with a 3.75-fold volume of an organic solvent containing chloroform and methanol at a 1:2 volume ratio. After vortexing, the mixture became turbid. The mixture was added with chloroform (1.2-fold of the volume for the reaction mix) and vortexed again. Clear phase separation was achieved after this step. The mix was added with Buffer B (1.2-fold of the volume for the reaction mix), vortexed, and centrifuged at 100 g for 5 min at RT. The bottom layer containing the chloroform phase was recovered and dried under a stream of argon. The product was then dissolved in 20 µL of 1:1 chloroform/methanol. Four microliters of the solution were applied to a Silica gel 60 F254 plate (Merk, 1.05554.001) that had pre-run in chloroform, along with standards and controls from reactions omitting the enzyme or the substrate. The plate was dried under a stream of argon and developed in a solvent system containing hexane: diethyl ether: acetic acid (70: 30: 3, vol: vol). The plate was air-dried, dipped into 10% phosphomolybdic acid (dissolved in ethanol), blotted with Kimwipe paper, and placed on a heater with a setting of ~250 °C. TLC images were taken using a smart phone.

## Generation of yeast strains

To generate the *sec13-1* mutant, a construct (pRS303-sec13m, Fig. S11b, i) was made as follows. A DNA fragment the ORF of *SEC13* (SGD ID S000004198) and the flanking region (662 bp at the 5′-end and 553 bp at the 3′-end) was amplified by PCR from the genomic DNA of *S. cerevisiae* YPH500 (*MATα ura3-52 lys2-801_amber ade2-101_ochre trp1-Δ63 his1-Δ200 leu2-Δ1*) cells. The fragment was Gibson assemblied (Cat. EG21202S, BestEnzymes Biotech) into the pRS303 vector (Cat. 77138, ATCC) between the *Spe*I and *Bam*HI site. Site-directed mutagenesis was performed to introduce nucleotides corresponding to the S244N mutation found in *sec13-1*, and the mutation was verified by DNA sequencing. (We would note that according to the nucleotide sequence in the original publication[48] describing

*sec13-1*, we annotate the mutation as S244N instead of S244K presented in the literature.)

For genome integration, three fragments were obtained by separate PCR reactions. The first fragment contains the ORF encoding the Sec13 S244N mutant with flanking regions (white box, Fig. S11b, i). The second fragment contains the *His3* ORF with flanking regions (gray box, Fig. S11b, i). The third fragment contains a region at the further 3′-end of *SEC13* (green box, Fig. S11b, i). All three fragments were amplified from this pRS303-sec13m (Fig. S11b, i). The three fragments were then assembled into a single fragment (Frag-sec13m, Fig. S11b, ii) by overlap PCR. This large fragment was purified from agarose gel and transformed into *S. cerevisiae* YPH500 competent cells by heat shock. Colonies grown on an agar plate with a synthetic defined medium lacking histidine were streaked onto two plates, which were then placed at 24 °C (permissive) and 37 °C (restrictive), respectively. Temperature-sensitive colonies were screened by sequencing, and positive colonies were used for further experiments.

To replace *yPGAP1* in the *sec13-1* cells with *cPGAP1* (genotype of the resulting cells: *sec13-1 ypgap1::cPGAP1*, Fig. 1c), the same procedure above was used except that the plasmid pRS304-cPGAP1 (Fig. S11c, i) was used to generate the fragment (Fig. S11c, ii) for transformation, and the colonies were only selected on medium lacking tryptophan before PCR identification and DNA sequencing. As a control, the procedure was repeated with *yPGAP1* using the plasmid pRS304-yPGAP1 (Fig. S11d) (the genotype of the resulting cells: *sec13-1 ypgap1::yPGAP1*, Fig. 1c). Note, the plasmid pRS304-yPGAP1 (Fig. S11d, i) was constructed first, to which the *yPGAP1* ORF was then replaced by the *cPGAP1* ORF by Gibson assembly to generate pRS304-cPGAP1 (Fig. S11c, i).

To generate yPGAP1(Bst1)-KO *S. cerevisiae* cells, the plasmid pRS304-ΔPGAP1 (Fig. S11e) was made as follows. A DNA fragment containing 250 bp downstream (3′-direction) of the *yPGAP1* ORF (Uniprot ID P4357 [https://www.uniprot.org/uniprotkb/P43571/entry]) was assembled with a second fragment containing a *Bam*HI restriction site followed by the 300 bp upstream (5′-direction) of the ORF by overlap PCR. The assembled PCR fragment was Gibson assemblied into the pRS304 vector at the multiple cloning site. The plasmid was linearized using *Bam*HI (Cat. ER0055, Thermo Fisher Scientific), transformed into YPH500 or *sec13-1* cells, and selected on a synthetic defined medium lacking tryptophan. Colonies were screened by PCR and verified by sequencing.

## Yeast growth assay

Colonies of different yeast strains were inoculated into 3 mL of media containing 2% glucose (Cat. 10010518, Sinopharm), 0.67% Yeast nitrogen base (YNB, Cat. 291930, BD Biosciences) and amino acids without omitting (YPH500) or omitting tryptophan (*ypgap1*), histidine (*sec13-1*), or histidine and tryptophan (*sec13-1 ypgap1*, *sec13-1 ypgap1::cPGAP1*, and *sec13-1 ypgap1::yPGAP1*) (brackets indicate the genotype of the cells used in the experiment in Fig. 1c) and cultured at 24 °C for two days with 250 r.p.m. in a shaking incubator. Cells were seeded into a fresh medium with an initial optical density at 600 nm ($OD_{600}$) of 0.15 and cultured separately in two incubators set at 24 °C (permissive temperature) and 35 °C (restrictive temperature), respectively. Cell growth was monitored by measuring the $OD_{600}$ at indicated time intervals on a SpectraMax M2e plate reader (Molecular Devices) using the software SoftMax Pro 7.1.2.

## Cryo-EM sample preparation and data acquisition

For the nanodisc sample cPGAP1$^{apo}$, a drop of 2.5 µL protein at 7.5 mg mL$^{-1}$ concentration was applied to glow-discharged Quantifoil Au R1.2/1.3 (300 mesh) grids. For the cPGAP1$^{H443N}$ and cPGAP1$^{S327A}$, a drop of 2.5 µL protein at 10 mg mL$^{-1}$ (H443N) or 14.9 mg mL$^{-1}$ (S327A) was applied to Quantifoil Au R1.2/1.3 (200 mesh) grids. The grids were loaded to a Vitrobot Mark IV (FEI) machine in a pre-cooled chamber

 

with a temperature of 8 °C and 100% humidity. Grids were blotted with a setting of time/force of 4.5 s/5 against two pieces of filter paper. After the blotting step, grids were rapidly plunged into pre-cooled liquid ethane.

Data for cPGAP1[apo] were collected with a Titan Krios cryo-electron microscope (Thermo Fisher Scientific) operated at 300 kV with a 50-µm condenser lens aperture, magnification at 105,000× (corresponding to a calibrated sampling of 0.832 Å per physical pixel), and a K3 direct electron device. Micrographs were collected automatically using the Serial EM software (version 3.8.0) with the detector operating in super-resolution mode at a recording rate of 10 raw frames per second and a total exposure time of 2 s, yielding 40 frames per movie stack and a total dose of 52 e⁻/Å². 

Data for cPGAP1[H443N] and cPGAP1[S327A] were collected with a Titan Krios G4 cryo-electron microscope (Thermo Fisher) operated at 300 kV with a 70 µm condenser lens aperture, spot size 4, magnification at 130,000× (corresponding to a calibrated sampling of 0.932 Å per physical pixel), and a Falcon 4i direct electron device equipped with a Selectris X energy filter operated with a 20 eV slit (Thermo Scientific). Movie stacks were collected automatically using the EPU software (Thermo Fisher) with the detector operating in counting mode at a total exposure time of 3.51 s, yielding 1080 frames per EER (electron event representation) movie and a total dose of 50 e⁻/Å².

### Cryo-EM data processing and model building

A total of 7283 movies (cPGAP1[apo] in nanodisc), 7948 movies (cPGAP1[H443N]), and 4310 movies (cPGAP1[S327A]) were collected and processed in RELION (v3.1)[76] and cryoSPARC (v4.2.0)[77]. Each electron-event representation movie of ~1080 frames were fractionated into 40 subgroups and beam-induced motion was corrected by RELION's own implementation. Exposure-weighted averages were then imported to cryoSPARC and the contrast transfer function parameters for each micrograph were estimated by CTFFIND4[78]. Particles were blob-picked and extracted with a box size of 270 pixels, and subjected to several rounds of 2D classification and heterogeneous refinement (3D classification) to remove contaminants or poor-quality particles. The good particles were then converted for Bayesian polishing in RELION, which was subsequently imported back to cryoSPARC for heterogeneous refinement. The final 2.84-Å cPGAP1[apo] map from 407,921 particles, 2.68-Å cPGAP1[H443N] map from 179,980 particles, and 2.66-Å cPGAP1[S327A] map from 353,585 particles were obtained by local refinement. Resolution of these maps was estimated internally in cryoSPARC by gold-standard Fourier shell correlation using the 0.143 criterion. Details for data processing are in Supplementary information (Figs. S5, S6, Table S1).

Models were built manually using Coot 0.9.6[79] and refined using Phenix 1.19.2-4158[80]. Structures were visualized using PyMOL 2.3.3 and ChimeraX1.1. For the GPI part, we could not distinguish the chain composition based on the cryo-EM density. Therefore, we used existing knowledge in the literature to build the acyl chains. Because cPGAP1 S327A did not show activity in HEK293 cells (Fig. 2g), the 2-acyl chain is expected to remain an unsaturated composition. Accordingly, we modeled it as a C20:4 (arachidonoyl) chain, as it is the most abundant form before fatty acid remodeling by PGAP3 and PGAP2[54].

Because cPGAP1 H443N retained apparent activity at a similar-to-wildtype level in HEK293 cells (Fig. 2e), a significant portion of TGP₂ is expected to go through the fatty acid remodeling by PGAP3 and PGAP2 (Fig. S1b)[2], generating mostly a C18:0 2-acyl chain[54] and anchoring to the cell surface. Upon solubilization, these processed TGP₂ may bind cPGAP1 to form an enzyme-product complex. In accord with this, the 2-acyl chain of TGP₂ in cPGAP1[H443N] was modeled as C18:0. However, we note the possibility that some TGP₂ remain bound to cPGAP1 in the ER membrane after the lipase reaction and thus are not further processed. This population is expected to mostly carry a C20:4 (arachidonoyl) chain[54].

### Reporting summary

Further information on research design is available in the Nature Portfolio Reporting Summary linked to this article.

## Data availability

The coordinates for the model generated in this study have been deposited in the PDB under accession code 8K9Q (cPGAP1[apo]), 8K9T (cPGAP1[S327A]), and 8K9R (cPGAP1[H443N]). The cryo-EM density maps generated in this study have been deposited in the Electron Microscopy Data Bank (EMDB) under accession code EMD-36995 (cPGAP1[apo]), EMD-36997 (cPGAP1[S327A]), and EMD-36996 (cPGAP1[H443N]). Uncropped images of Fig. 1e and tabular data for Figs. 1c, 1g, 3e, 4c, 5b and 5e are provided in the Source Data file. The uncropped images of Figs. S4c, S4e, S4f and S4h are provided in Fig. S13. Source data are provided as a Source Data file. Source data are provided with this paper.

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

## Acknowledgements

We thank the staff members of Core Facilities of Molecular Biology and Cell Biology in Shanghai Institute of Biochemistry and Cell Biology, the Center of Cryo-EM of Fudan University, the NCPSS Electron Microscopy facility for technical support and assistance. This work has been supported by the National Natural Science Foundation of China (82151215, D.L.; 32171194 & 32371256, Q.Q.; 32201000, T.L.), the Ministry of Science and Technology of China (2023YFA0915000 to Q.Q.), the Strategic Priority Research Program of CAS (XDB37020204), the China Post-doctoral Science Foundation (2022M720805, Z.Z.), the Heye Scholarship Program (T.L.), Science and Technology Commission of Shanghai Municipality (22ZR1468300, D.L.), the Shanghai Post-doctoral Excellence Program (2021378, T.L.), and the start-up funds from Shanghai Stomatological Hospital & School of Stomatology, Fudan University (Q.Q.). We thank Prof. Jinqiu Zhou and Dr. Qianxi Li at our institute for providing strains, plasmids and technical guidance for yeast genetics.

## Author contributions

J.H. purified protein and performed assays. T.L. purified complexes and performed assays. Y.C. prepared and screened cryo-EM grids and collected cryo-EM data with assistance from Z.Zhou. Y.X. established initial protein production protocols. Q.Q. and Z.Zhu. processed cryo-EM data. W.G. helped with molecular cloning and protein expression. D.L. designed assays. D.L. and Q.Q. oversaw the project. D.L. wrote the manuscript with input from Q.Q., T.L., J.H. and Y.X.

## Competing interests

The authors declare no competing interests.
