## [Peer Review File · Nature Communications]

Molecular basis of the inositol deacylase PGAP1 involved in quality control of GPI-AP biogenesisReviewers' Comments:

Reviewer #1:

Remarks to the Author:

The manuscript by Hong et al. elucidated the molecular structure of the GPI inositol deacylase, PGAP1. The authors identified a stable PGAP1 from *Chaetomium thermophilum* and worked on the enzymatic and structural analysis. In the enzymatic analysis, the authors developed an original assay system combining size exclusion chromatography and PIPLC treatment, and studied the effect of amino acid residue substitution on GPI inositol deacylation based on structural analysis. In addition, structural analysis using cryo-electromicroscopy revealed the holostructure of WT PGAP1 and the complex structure with GPI-anchored proteins in the bound state at H443N and S327A. They proposed mechanisms of substrate recognition and enzymatic reaction. This is the first result revealing the processing mechanism of GPI-anchor and should be highly appreciated. The reviewer made several comments below and asked the authors to respond appropriately.

1) From the structural results, the authors proposed that cholesterol is involved in the release of deacylated GPI-APs from the protein complex. If the importance of cholesterol is shown in the enzymatic analysis, this is a very impressive result, revealed for the first time by the structure. Using the PGAP1 assay system developed by the authors, the reviewer asks the authors to confirm whether PGAP1 activity is altered in the presence or absence of cholesterol.

2) In the protein alignment of Fig. S2, cPGAP1 has an additional sequence at the N-terminus (about 130 aa) compared to human PGAP1. Have the authors analysed the full-length cPGAP1 structure by cryo-EM? The reviewer could not find any mention in the text whether it is the full-length or truncated structure. Please describe this. If the authors analysed the full length, what structure was observed at this N-terminal region?

Also, in the complementation experiments with mammalian cells or yeast, do the authors express the full length or not? The authors should explain the details of the cPGAP1 used in the analysis.

3) The authors showed that the second EtNP was not detected in the complex structure of cPGAP1 and the GPI anchor. One possibility, according to the authors, is that the second EtNP is flexible. Since the authors co-expressed less-active PGAP1 and TGP in PGAP1-KO cells, it is possible that the second EtNP has already been cleaved by PGAP5 because TGP remains in the ER for a long time. The best way to show this is to use PGAP1-PGAP5 double KO cells. But at least this possibility should be discussed if the authors agree that it is reasonable.

4) The authors have shown that there are two jelly roll structures in the structure of cPGAP1. From the sequence alignment in Fig. S2, the region appears to have less homology between human, yeast and *Chaetomium* compared to the lipase domain. The reviewer recommends modelling to analyse whether the similar structure exists in human and yeast PGAP1.

Minor points:

Line 69: mammal -> mammalian

Line 76: "and lipid raft association of GPI-APs."; In PGAP3-mutant cells, GPI-APs having an unsaturated fatty acid are not associated with lipid rafts. However, it is still not clear whether tri-acylated GPI-APs in PGAP1-KO cells are associated with lipid rafts or not. The reviewer recommends to delete the sentence.

Line 85: Add the Reference 49 as well, here.

Line 175: phosphoinositol -> phosphatidylinositol

Fig.S4a: The position of TGP and StrepII/Flag-tag is different between (i) and (ii). Correct the chimeric GPI-AP construct.

Line 488: GPI-Aps -> GPI-APs

Line 569, 571: GpI-APs -> GPI-APs

Reviewer #2:

Remarks to the Author:

GPI-anchored proteins (GPI-APs) expressed on the cell surface play various important roles in eukaryotic organisms. Nascent GPI-APs are generated in the ER by transfer of preassembled GPI precursor proteins, and then become mature GPI-APs after several maturation reactions. The first maturation reaction is removal of acyl chain from the inositol ring, inositol-deacylation, that is critical for quality control of GPI-APs. An ER-resident multi-transmembrane enzyme PGAP1 catalyzes inositol-deacylation. Structure and reaction mechanism of PGAP1 have been unclear. Li, Qu and their colleagues studied molecular basis of inositol-deacylation using the enzyme from a thermophilic fungus, termed cPGAP1. Presumably thanks to stable nature of the enzyme from the thermophilic fungus, authors were able to determine three structures of cPGAP1 with or without GPI-AP. For enzymatic reaction characterization in vitro and structural analysis of the deacylation reaction by cryo-EM, authors generated a recombinant artificial GPI-AP substrate, TGP3, bearing an inositol-linked acyl-chain in human HEK293 cells. Therefore, the study used a combination of fungal enzyme and human-cell-derived substrate.

Authors demonstrated that cPGAP1 consists of a lipase domain, transmembrane domain with 10 transmembrane helices, and two jelly-roll domains; that cPGAP1 holds acyl chains of GPI-AP in a guitar-shaped cavity made by the lipase domain and several transmembrane helices; that 2-acyl chain of PI is inserted into a deep pocket by 5~6 carbons; and that glycan moiety of GPI has abundant interactions with cPGAP1 in the lumen. Further, authors found that GPI-APs buried in two forms of GPI-AP-associated cPGAP1 were in fact products, acyl chain being free from the main part of GPI-AP and that orientation of the main parts were different in two forms. Comparing two forms, authors proposed a mechanism of product release.

Overall, this study advances our understanding of mechanisms of GPI-APs' maturation after attachment of GPI to proteins. There are several points to be addressed by authors for clarity and accurate interpretation of the data.

Major points.

1, Page 9. For quantitative inositol-deacylase assay, authors prepared TGP3 substrate in PGAP1-KO HEK293 cells. With TGP3, cPGAP1 showed surprisingly slow activity and rather high K_m . Based on the previous data about glycan structures of GPI-APs derived from HEK293 cells, I guess that purified TGP3 molecules were not homogeneous but contained Man3-form as a major form and Man4-form as a minor form. cPGAP1 co-expressed with TGP3 bound Man4-form where Man4 made three H-bonds with cPGAP1. It seems likely that Man4-form is a natural substrate of cPGAP1 because Man4-form is predominant in fungal GPI. It is therefore possible that only minor Man4-form was used by cPGAP1 and hence V_{max} and K_m were under/over-estimated. One way to make Man4-form only substrate for accurate measurement of kinetics parameters of cPGAP1 is to use PGAP1-KO and PIGZ-transfected HEK293.

2, Page 19, Figure 4d and Figure S10.

cPGAP2-associated TGP2 had 1-alkyl C18 chain and 2-acyl C18 chain. Is it possible to comment whether these chains contain any unsaturated bond? Also, is this the only chain composition found? I guess that TGP3 derived from HEK293 cells had heterogenous chain compositions, especially including those with polyunsaturated 2-acyl chains and those with very long chain such as C20 and C22.

The 2-acyl chain with C18 was inserted into a deep pocket by 5~6 carbon atoms. It would be useful if authors comment whether TGP3 bearing C20 or C22 is expected to be accommodated in or excluded

from cPGAP1.

Other points.

- 1, Figure S4a. Orientations of Strep tag and TGP are reversed in panels i and ii. This needs to be fixed.
- 2, Page 3, line 48. VSG stands for "variant surface glycoprotein" not "variable".
- 3, Page 3, line 57. EtNP should be defined.
- 4, Page 21, line 500. "Shallow groove" not "shadow groove"?
- 5, Ref#3. Peter, O. should be Orlean, P.

Reviewer #3:

Remarks to the Author:

In manuscript titled "Molecular basis of the inositol deacylase PGAP1 involved in quality control of GPI-AP biogenesis", Jingjing Hong et al reported the cryo-EM structures of wildtype and two mutants of PGAP1. Nowadays, to present the structure is not so exciting due to the development of the cryo-EM technology. In this manuscript, they also reveal a serine hydrolase-type catalysis with atypical features and suggest a unique mechanism for product release involving a cholesterol-mediated "drawing compass" movement of GPI-APs. This mechanism is interesting to me, but I think there are still some issues that need to be addressed.

Major Comment #1: Depending on the data processing, I remain doubtful about the structural heterogeneity in the particles resulting in the so-called hydrolase-type TGPx, therefore it is necessary to carefully check the data processing to exclude this problem.

Major Comment #2: There are only three structures, the proposed mechanism for catalysis and product release seems over-interpreted. Modification of some claims may be more appropriate.

Minor Comments:

Are there any other differences between these three structures, and if so, do they influence the function of the protein.

Overall, I recommend that this manuscript could be published after addressing the major and minor concerns.

Response to reviewer comments

Reviewer #1

The manuscript by Hong et al. elucidated the molecular structure of the GPI inositol deacylase, PGAP1. The authors identified a stable PGAP1 from *Chaetomium thermophilum* and worked on the enzymatic and structural analysis. In the enzymatic analysis, the authors developed an original assay system combining size exclusion chromatography and PIPLC treatment, and studied the effect of amino acid residue substitution on GPI inositol deacylation based on structural analysis. In addition, structural analysis using cryo-electromicroscopy revealed the holostructure of WT PGAP1 and the complex structure with GPI-anchored proteins in the bound state at H443N and S327A. They proposed mechanisms of substrate recognition and enzymatic reaction. This is the first result revealing the processing mechanism of GPI-anchor and should be highly appreciated. The reviewer made several comments below and asked the authors to respond appropriately.

We thank the reviewer for the supportive comments.

1) From the structural results, the authors proposed that cholesterol is involved in the release of deacylated GPI-APs from the protein complex. If the importance of cholesterol is shown in the enzymatic analysis, this is a very impressive result, revealed for the first time by the structure. Using the PGAP1 assay system developed by the authors, the reviewer asks the authors to confirm whether PGAP1 activity is altered in the presence or absence of cholesterol.

We thank the reviewer for the insightful suggestion.

We have conducted assays under varying concentrations of cholesterol. Intended as a control, we also performed assays with the cholesteryl hemisuccinate (CHS). The results showed that cholesterol was inhibitory to cPGAP1, reducing its activity by approximately a half at 5 mol%. CHS was less inhibitory. We propose that cholesterol inhibits activity by suppressing substrate binding, as it occupies the position of the 1-alkyl chain.

Accordingly, we have revised the text and added **Fig. 5e** to the revised manuscript, as quoted below.

“A proposed mechanism for substrate entrance and product release

Despite the overall structure similarity, the two product-bound structures (cPGAP1^{H443N} and cPGAP1^{S327A}) exhibit notable differences in the positioning of the TGP₂ *sn*-1 alkyl chain, offering insights into the processes of substrate entrance and product release. In cPGAP1^{H443N}, the 1-alkyl chain tightly associates with the enzyme (**Fig. 5d**). Conversely, in cPGAP1^{S327A}, it undergoes a ~45° “drawing compass” rotation, shifting outward by 6 Å and detaching from the enzyme (**Fig. 5d**). This suggests the possibility that the substrate initially anchors to the enzyme by inserting the 2-acyl into the side pocket (**Fig. 4d**), resembling a compass needle (**Fig. 5d**). Subsequently, the 1-alkyl chain, analogous to a compass pencil, swings towards the

enzyme (**Fig. 5d**). During product release, GPI-AP₂ may undergo a similar motion but in the reverse direction (**Fig. 5d**), exposing the 1-alkyl chain to the bulk membrane. This exposure may facilitate downstream enzymes like PGAP5 to bind the alkyl chain (along with the exposed glycan), potentially aiding in substrate channeling.

Given that PLM remains bound to cPGAP1 during purification and is enclosed by GPI-AP₂ in the guitar-shaped cavity (**Fig. 4b**), it is likely that the fatty acid is released after GPI-AP₂.

In cPGAP1^{S327A}, a blob of density fitting well for a cholesterol molecule (**Fig. 2f**) occupies the 1-alkyl position of the TGP₂ in cPGAP1^{H443N} (**Fig. 5d**). To explore possible functional consequences of this observation, we conducted enzymatic assays under varying cholesterol concentrations while maintaining a constant detergent concentration. Cholesterol had no discernible effect on cPGAP1 activity at 0.08 mol% or lower concentrations. However, inhibition became evident at 0.33 mol%, reducing cPGAP1 activity by approximately half at 5 mol% (**Fig. 5e**), a concentration (in relation to total lipids) found in the ER membranes of mammalian cells⁶⁰. This effect appeared somewhat specific, as the hemisuccinate derivative CHS showed lesser inhibition (**Fig. 5e**). These results suggest the functional significance of the binding between the 1-alkyl moiety and enzyme, and a regulatory role of cholesterol in substrate binding, assuming a causative relationship between cholesterol binding at this site and the observed inhibition.”

Fig. 5 | Mechanisms for catalysis and product release.

e Effect of CHL (triangle) and CHS (circle) on cPGAP1 activity. Average \pm s.d. from three independent experiments are plotted. Statistical analyses were performed using one-way ANOVA followed by Dunnett's multiple comparisons test. Source data of **b** and **e** are provided as a Source Data file.

2) In the protein alignment of Fig. S2, cPGAP1 has an additional sequence at the N-terminus (about 130 aa) compared to human PGAP1. Have the authors analysed the full-length cPGAP1 structure by cryo-EM? The reviewer could not find any mention in the text whether it is the full-length or truncated structure. Please describe this.

The full-length protein was used. The information has been added in Line 151, 223-224,

and 583-584 in the revised manuscript.

The model completeness information has also been included in the revised **Fig. S7a**, where unobserved regions are indicated by dashed lines.

Fig. S7 | Topology and the jelly-roll domains of cPGAP1. a Topology of cPGAP1. ... Dashed lines mark unresolved regions, with the starting and ending residues indicated.

If the authors analysed the full length, what structure was observed at this N-terminal region?

We did not observe the N-terminal region. This region is presumed to be disordered. The information has been added to **Fig. S2** for clarity, as quoted below:

“**Fig. S2 | Sequence alignment of PGAP1 orthologs. a** Sequence alignment of PGAP1 from *Chaetomium thermophilum*, human, and *Saccharomyces cerevisiae*. ... The first 130 residues of cPGAP1 and the first 44 residues of yPGAP1, which are predicted to be disordered, are not shown due to their poor sequence homology.”

Also, in the complementation experiments with mammalian cells or yeast, do the authors express the full length or not? The authors should explain the details of the cPGAP1 used in the analysis.

We used full-length cPGAP1 constructs for the complementation experiments. This information has been added in Line 151 of the revised manuscript.

We have also included the following sentence in the Method section:

“All PGAP1 constructs used in this study contain the full-length protein unless otherwise

specified, such as the truncation constructs for the jelly-roll domains.”

3) The authors showed that the second EtNP was not detected in the complex structure of cPGAP1 and the GPI anchor. One possibility, according to the authors, is that the second EtNP is flexible. Since the authors co-expressed less-active PGAP1 and TGP in PGAP1-KO cells, it is possible that the second EtNP has already been cleaved by PGAP5 because TGP remains in the ER for a long time. The best way to show this is to use PGAP1-PGAP5 double KO cells. But at least this possibility should be discussed if the authors agree that it is reasonable.

We agree with this possibility. We have modified the text as follows:

“Despite the well-defined density of the GPI-anchor, there was no evidence for the EtNP2. EtNP2 may exist in a highly mobile state due to a lack of interactions with cPGAP1, as suggested by the fact that the Man2 6-hydroxyl (where EtNP2 is attached) is exposed to the bulk solvents (**Fig. 4a**). This exposure **implies** accessibility to PGAP5, an ER membrane-residing remodelase that cleaves EtNP2 and primes GPI-APs for ER export⁵⁷. **Extending this idea, it is conceivable that EtNP2 may have been removed, especially given the presumed long retention of TGP_{2/3} in the ER membrane, attributed to the compromised activity of cPGAP1 H443N and cPGAP1 S327A.**”

4) The authors have shown that there are two jelly roll structures in the structure of cPGAP1. From the sequence alignment in Fig. S2, the region appears to have less homology between human, yeast and Chaetomium compared to the lipase domain. The reviewer recommends modelling to analyse whether the similar structure exists in human and yeast PGAP1.

We thank the reviewer for the suggestion.

The three homologs indeed share low sequence homology. However they all contain two jelly-roll subdomains arranged similarly as cPGAP1.

We have included a new figure (**Fig. S7b**) to show the similarities, as quoted below.

“The two jelly-roll subdomains, which also exist in the predicted⁵³ hPGAP1 and yPGAP1 models (Fig. S7b), pack against each other through a network of hydrophobic interactions, mostly of aromatic residues (Fig. 3c)...”

Fig. S7 | Topology and the jelly-roll domains of cPGAP1.

b Comparison of the jelly-roll domains between cPGAP1 (top), hPGAP1 (bottom left), and yPGAP1 (bottom right).

Minor points:

Line 69: mammal -> mammalian

Done.

Line 76: “and lipid raft association of GPI-APs.”; In PGAP3-mutant cells, GPI-APs having an unsaturated fatty acid are not associated with lipid rafts. However, it is still not clear whether tri-acylated GPI-APs in PGAP1-KO cells are associated with lipid rafts or not. The reviewer recommends to delete the sentence.

Point taken.

Line 85: Add the Reference 49 as well, here.

Done.

Line 175: phosphoinositol -> phosphatidylinositol

Done.

Fig.S4a: The position of TGP and StrepII/Flag-tag is different between (i) and (ii). Correct the chimeric GPI-AP construct.

Done

Line 488: GPI-Aps -> GPI-APs

Done

Line 569, 571: GpI-APs -> GPI-APs

Done

We thank the reviewer for the careful reading.

Reviewer #2

GPI-anchored proteins (GPI-APs) expressed on the cell surface play various important roles in eukaryotic organisms. Nascent GPI-APs are generated in the ER by transfer of preassembled GPI to precursor proteins, and then become mature GPI-APs after several maturation reactions. The first maturation reaction is removal of acyl chain from the inositol ring, inositol-deacylation, that is critical for quality control of GPI-APs. An ER-resident multi-transmembrane enzyme PGAP1 catalyzes inositol-deacylation. Structure and reaction mechanism of PGAP1 have been unclear. Li, Qu and their colleagues studied molecular basis of inositol-deacylation using the enzyme from a thermophilic fungus, termed cPGAP1. Presumably thanks to stable nature of the enzyme from the thermophilic fungus, authors were able to determine three structures of cPGAP1 with or without GPI-AP. For enzymatic reaction characterization in vitro and structural analysis of the deacylation reaction by cryo-EM, authors generated a recombinant artificial GPI-AP substrate, TGP3, bearing an inositol-linked acyl-chain in human HEK293 cells. Therefore, the study used a combination of fungal enzyme and human-cell-derived substrate.

Authors demonstrated that cPGAP1 consists of a lipase domain, transmembrane domain with 10 transmembrane helices, and two jelly-roll domains; that cPGAP1 holds acyl chains of GPI-AP in a guitar-shaped cavity made by the lipase domain and several transmembrane helices; that 2-acyl chain of PI is inserted into a deep pocket by 5~6 carbons; and that glycan moiety of GPI has abundant interactions with cPGAP1 in the lumen. Further, authors found that GPI-APs buried in two forms of GPI-AP-associated cPGAP1 were in fact products, acyl chain being free from the main part of GPI-AP and that orientation of the main parts were different in two forms. Comparing two forms, authors proposed a mechanism of product release.

Overall, this study advances our understanding of mechanisms of GPI-APs' maturation after attachment of GPI to proteins. There are several points to be addressed by authors for clarity and accurate interpretation of the data.

We thank the reviewer for the supportive comments.

Major points.

1, Page 9. For quantitative inositol-deacylase assay, authors prepared TGP3 substrate in PGAP1-KO HEK293 cells. With TGP3, cPGAP1 showed surprisingly slow activity and rather high K_m . Based on the previous data about glycan structures of GPI-APs derived from HEK293 cells, I guess that purified TGP3 molecules were not homogeneous but contained Man3-form as a major form and Man4-form as a minor form. cPGAP1 co-expressed with TGP3 bound Man4-form where Man4 made three H-bonds with cPGAP1. It seems likely that Man4-form is a natural substrate of cPGAP1 because Man4-form is predominant in fungal GPI. It is therefore possible that only minor Man4-form was used by cPGAP1 and hence V_{max} and K_m were under/over-estimated. One way to make Man4-form only substrate for accurate measurement of kinetics parameters of cPGAP1 is to use PGAP1-KO and PIGZ-transfected HEK293.

We thank the reviewer for bringing up this insightful aspect for discussion.

We have tried co-expressing PIGZ and TGP₃ in PGAP1-KO HEK293 cells to obtain Man4-modified TGP₃. However, the expression level of PIGZ was very low (undetectable by in-gel fluorescence which has a sensitivity of ~1 ng of protein). A preliminary assay using TGP₃ from PIGZ-transfected cells did not show higher activity compared to TGP₃ from non-transfected cells (**Fig. R1**). Because the low expression level of PIGZ, the results were inconclusive. Future optimizations are required to clarify this point.

Fig. R1. Preliminary investigation of the effect of Man4 on cPGAP1 activity. **a, b** FSEC assay of cPGAP1 using TGP₃ purified from hPGAP1-KO cells transfected with **(a)** or without **(b)** PIGZ. **i-iii** indicate three replicates. **c** Schematic of data analysis for cPGAP1 activity. **d** Bar graph of the results from **a** and **b**. We note the higher background of the TGP₀ peak in **a**. Usually, we remove this peak by gel filtration (as in **b**). However, due to its low yield (1/6 compared to cells without PIGZ transfection), we did not perform gel filtration for the preliminary assay.

Although we could not answer this question within the time frame of the revision, we consider this a very interesting point for discussion and have added the following to the revised manuscript:

“...Alternatively, the low apparent activity may be linked to the species-specific glycan preferences. Thus, the Man4 modification is required for GPI-AP biogenesis in *S. cerevisiae*, and by reasonable extension, in *C. thermophilum* which is also a fungus. Although the involvement of the PGAP1 step in the Man4 requirement is unknown, the product-bound structures of TGP₃ trapped in our study contained the Man4 moiety, despite its scarcity in mammalian cells⁶³. In addition, the purification yield of cPGAP1 H443N/S327A-TGP3 complex was approximately 10% at the second affinity chromatography step, implying a heterogeneous nature of TGP₃ with varying suitability as cPGAP1 substrates. Supporting this speculation, our prolonged enzymatic assays using saturating wild-type enzyme only achieved ~90% completion. Moreover, Man4 formed three H-bonds with cPGAP1 (**Fig. 4e**), with noteworthy observations that related mutations increased activity (**Table S2**). Future experiments using cell lines overexpressing PIGZ (the enzyme responsible for Man4 modification)² or PIGZ-KO cells may help clarify whether Man4-modification increases cPGAP1 activity. A yet another hypothesis is that PGAP1 might exhibit higher activity in native membranes, a possibility requiring a new assay format to test.”

2, Page 19, Figure 4d and Figure S10.

cPGAP2-associated TGP2 had 1-alkyl C18 chain and 2-acyl C18 chain. Is it possible to comment whether these chains contain any unsaturated bond?

We thank the reviewer for raising this question.

We could not distinguish the chain composition based on the cryo-EM density. Therefore, we used existing knowledge to build the acyl chains.

Because cPGAP1 H443N retained activity in HEK293 cells (**Fig. 2e**), TGP₂ is expected to go through the fatty acid remodeling by PGAP3 and PGAP2, generating mostly a C18:0 2-acyl chain. Accordingly, the 2-acyl chain of TGP₂ in cPGAP1^{H443N} was modeled as C18:0 (**Fig. R2a**).

Conversely, because cPGAP1 S327A did not show activity in HEK293 cells (**Fig. 2g**), the 2-acyl chain is expected to remain an unsaturated composition. Accordingly, we modeled it as a C20:4 (arachidonoyl) chain (**Fig. R2b**), as it is the most abundant form found in mammalian cells.

Fig. R2. Density and fitting of the 2-acyl moiety of TGP₂. **a** cPGAP1^{H443N}. **b** cPGAP1^{S327A}. Relevant parts are labeled for orientation purposes.

This information has been added to the Methods section of the revised manuscript with appropriate references:

“Models were built manually using Coot⁷⁹ and refined using Phenix⁸⁰. For the GPI part, we could not distinguish the chain composition based on the cryo-EM density. Therefore, we used existing knowledge in the literature to build the acyl chains. Because cPGAP1 H443N retained activity in HEK293 cells (**Fig. 2e**), TGP₂ is expected to go through the fatty acid remodeling by PGAP3 and PGAP2 (**Fig. S1b**)², generating mostly a C18:0 2-acyl chain⁵⁴. Accordingly, the 2-acyl chain of TGP₂ in cPGAP1^{H443N} was modeled as C18:0. Conversely, because cPGAP1 S327A did not show activity in HEK293 cells (**Fig. 2g**), the 2-acyl chain is expected to remain an unsaturated composition. Accordingly, we modeled it as a C20:4 (arachidonoyl) chain, as it is the most abundant form found in mammalian cells⁵⁴.”

Also, is this the only chain composition found?

For the same reasons mentioned above, we could not tell the chain composition based on the cryo-EM density.

I guess that TGP3 derived from HEK293 cells had heterogenous chain compositions, especially including those with polyunsaturated 2-acyl chains and those with very long chain such as C20 and C22.

Agreed. We chose to model the most abundant chain types based on the current knowledge and the rationale mentioned above.

The 2-acyl chain with C18 was inserted into a deep pocket by 5~6 carbon atoms. It would be useful if authors comment whether TGP3 bearing C20 or C22 is expected to be accommodated in or excluded from cPGAP1.

As mentioned above, both C18:0 and C20:4 acyl chains can be accommodated in this pocket.

We have added the following sentences (blue) in the revised manuscript:

“In contrast to the constrained binding of the inositol acyl, the diradyl moiety has more freedom in the chamber, showing some level of flexibility as indicated by its less-defined density (**Fig. 2c**). Typically, in mammalian cells, the 2-acyl chain of GPI-AP₃ consists of unsaturated fatty acids with 16 to 22 carbons⁵⁵. In the two product-bound structures, TGP₂ was built with a C18:0 (cPGAP1^{H443N}) and a C20:4 (cPGAP1^{S327A}) 2-acyl chain (see Methods) without causing clashes. Moreover, the chamber appeared spacious enough to accommodate a C22 2-acyl chain. Consistently, introducing bulky residues at several positions in the chamber (V362F, V925F, I932F) had no impact on or even increased the deacylase activity (**Fig. 4c**).”

Other points.

1, Figure S4a. Orientations of Strep tag and TGP are reversed in panels i and ii. This needs to be fixed.

Done

2, Page 3, line 48. VSG stands for “variant surface glycoprotein” not “variable”.

Done

3, Page 3, line 57. EtNP should be defined.

Done

4, Page 21, line 500. “Shallow groove” not “shadow groove”?

Done

5, Ref#3. Peter, O. should be Orlean, P.

Done

We thank the reviewer for the careful reading of our manuscript.

Reviewer #3

In manuscript titled “Molecular basis of the inositol deacylase PGAP1 involved in quality control of GPI-AP biogenesis”, Jingjing Hong et al reported the cryo-EM structures of wildtype and two mutants of PGAP1. Nowadays, to present the structure is not so exciting due to the development of the cryo-EM technology. In this manuscript, they also reveal a serine hydrolase-type catalysis with atypical features and suggest a unique mechanism for product release involving a cholesterol-mediated "drawing compass" movement of GPI-APs. This mechanism is interesting to me, but I think there are still some issues that need to be addressed.

We thank the reviewer for evaluating our work.

Major Comment #1: Depending on the data processing, I remain doubtful about the structural heterogeneity in the particles resulting in the so-called hydrolase-type TGP_x, therefore it is necessary to carefully check the data processing to exclude this problem.

We thank the reviewer for raising this very important question.

To assess potential structural heterogeneity, we conducted focused 3D classification without alignment in RELION, followed by refinement in cryoSPARC. Specifically, we generated a mask covering the core map regions around where TGP₂ was modeled, and performed 3D classification with 4 subclasses and a regularization T value of 40. This yielded three relatively ‘good’ classes (Class 1, Class 2, Class 3) and one ‘poor’ class (Class 4) (**Fig. R3**). To mitigate model bias, we performed *ab initio* reconstruction for each subclass. Subsequent non-uniform refinement and local refinement produced three high-quality maps, with 40,089 particles for Class 1 (2.95 Å), 51,226 for Class 2 (2.98 Å), and 145,207 for Class 3 (3.00 Å).

The cryo-EM density near the palmitic acid (PLM) and the inositol (Ino) of TGP₂ was then examined. As shown in **Fig. R4**, the density between PLM and Ino was discontinuous. Notably, no densities were observed extending from the C2 of Ino, where PLM is covalently attached in TGP₃. Therefore, we concluded that the majority of GPI-anchored TGP in our sample was TGP₂. If TGP₃ was present, its existence was not significant enough to manifest as a distinct class in the data processing.

Fig. R3. Data processing flowchart of cPGAP1^{S327A}.

Fig. R4. Cryo-EM density near the PLM and Ino region. a-c, Class 1, 2, and 3, respectively. Magenta arrow indicates the C2 of Ino, where PLM is covalently attached in TGP₃.

Major Comment #2: There are only three structures, the proposed mechanism for catalysis and product release seems over-interpreted. Modification of some claims may be more appropriate.

We thank the reviewer for the advice to avoid over claiming.

We have modified the paragraph by adding the blue texts as below:

“Based on the structural analysis and mutagenesis data, we propose a mechanism similar to triad lipases and serine proteases⁵⁹ (**Fig. 5c**). Asp407 increases the pK_a of His443, which in turn activates the Ser327 hydroxyl. On the other hand, the substrate GPI-AP₃, as illustrated in manually fitted and superimposed models in **Fig. S9**, is positioned with the scissile ester bond near the oxyanion hole, with the carboxyl of the superposed PLM and GPI-AP₃ sandwiched between the backbone amides of Met328 and Asn230 (**Fig. 5a**). Akin to triad hydrolyases⁵⁹, this configuration further polarizes and activates the C=O bond, facilitating the nucleophilic attack of the carbonyl carbon (of the manually fitted GPI-AP₃) by the primed Ser327 (**Fig. 5c, i**), which is 3.0-Å away (measured from manually fitted and superimposed models) (**Fig. 5a**). This results in the formation of an enzyme-substrate ether bond at the expense of the collapse of the carbonyl (**Fig. 5c, ii**). The product GPI-AP₂ and an acylated intermediate are generated through the regeneration of the carbonyl (**Fig. 5c, ii**). Subsequently, a water molecule activated by His443 attacks (**Fig. 5c, iii**) and hydrolyzes the intermediate through another cycle of collapse and regeneration of the carbonyl moiety (**Fig. 5c, iv**), producing the product fatty acid (FA) (**Fig. 5c, v**). “

We have also added the following texts in the figure legends of Fig. 5c.

“...c The serine hydrolase-type mechanism. The structures from this work only captured the state v. The rest of the steps were speculated based on existing knowledge on triad proteases.”

Minor Comments:

Are there any other differences between these three structures, and if so, do they influence the function of the protein.

We did not observe conformational changes between the three structures that are significant enough for us to make functional sense of.

The comparison of the structures is shown below (**Fig. R5**).

Fig. R5. Pairwise structural comparison between cPGAP1 apo (green), cPGAP1 H443N-TGP₂ (magenta), and cPGAP1 S327A-TGP₂ (cyan).

Overall, I recommend that this manuscript could be published after addressing the major and minor concerns.

We thank the reviewer for the supportive comments.

Reviewers' Comments:

Reviewer #1:

Remarks to the Author:

The authors addressed the reviewers' comments and adequately revised the manuscript. The reviewer recommends to publish the paper in Nature Communications.

Reviewer #2:

Remarks to the Author:

Authors addressed my points and revised the manuscript properly except one small point.

Concerning my major point 2, authors explained a basis of their choice of C18:0 2-acyl chain in modeling. They chose C18:0 because it is a predominant chain after fatty acid remodeling. Although it is correct that C18:0 is predominant, fatty acid remodeling occurs in the Golgi apparatus after inositol-deacylation by PGAP1 in the ER. It is therefore unlikely or a rare case that PGAP1 handles GPI-AP bearing C18:0 2-acyl chain. The corresponding description (lines1085-1088) needs to be modified.

Otherwise, I think the revised manuscript provides interesting data that greatly advance our understanding of mechanistic basis of how PGAP1 handles wide range of GPI-AP substrates. I support publication of the revised manuscript in Nature Communications.

Reviewer #3:

Remarks to the Author:

I have no more concerns, the authors have answered most of my questions.

REVIEWERS' COMMENTS

Reviewer #1 (Remarks to the Author):

The authors addressed the reviewers' comments and adequately revised the manuscript. The reviewer recommends to publish the paper in Nature Communications.

We thank the reviewer for the support.

Reviewer #2 (Remarks to the Author):

Authors addressed my points and revised the manuscript properly except one small point.

Concerning my major point 2, authors explained a basis of their choice of C18:0 2-acyl chain in modeling. They chose C18:0 because it is a predominant chain after fatty acid remodeling. Although it is correct that C18:0 is predominant, fatty acid remodeling occurs in the Golgi apparatus after inositol-deacylation by PGAP1 in the ER. It is therefore unlikely or a rare case that PGAP1 handles GPI-AP bearing C18:0 2-acyl chain. The corresponding description (lines 1085-1088) needs to be modified.

We thank the reviewer for the insightful discussion.

The reviewer's question concerns cPGAP1 H443N. Although only showing 0.4% activity in our biochemical assay, it displayed an activity level somewhat similar to the wild-type when overexpressed in cells (Fig. 2e). Therefore, we speculate that some TGP₂ in cPGAP1 H443N went through normal fatty acid remodelling by PGAP3/PGAP2, and thus carrying a C18:0 chain and anchoring to the plasma membrane. Upon solubilisation, they may bind to cPGAP1 H443N and be purified as the enzyme-product complex.

Having said this, we also acknowledge the following possibility. Some TGP₂ may remain bound to cPGAP1 H443N after the deacylation. They would carry a C20:4 chain.

We have modified the paragraph as below:

“Because cPGAP1 H443N retained apparent activity at a similar-to-wildtype level in HEK293 cells (**Fig. 2e**), a significant portion of TGP₂ is expected to go through the fatty acid remodeling by PGAP3 and PGAP2 (**Fig. S1b**)², generating mostly a C18:0 2-acyl chain⁵⁴ and anchoring to the cell surface. Upon solubilization, these processed TGP₂ may bind cPGAP1 to form an enzyme-product complex. In accord with this, the 2-acyl chain of TGP₂ in cPGAP1^{H443N} was modeled as C18:0. However, we note the possibility that some TGP₂ remain bound to cPGAP1 in the ER membrane after the lipase reaction and thus are not further processed. This population is expected to mostly carry a C20:4 (arachidonoyl) chain⁵⁴. ”

Otherwise, I think the revised manuscript provides interesting data that greatly advance our understanding of mechanistic basis of how PGAP1 handles wide range of GPI-AP substrates. I support publication of the revised manuscript in Nature Communications.

We thank the reviewer for the support and the constructive comments on our manuscript.

Reviewer #3 (Remarks to the Author):

I have no more concerns, the authors have answered most of my questions.

We thank the reviewer for the previous comments in improving our manuscript.